# REPA Works Until It Doesn't: Early-Stopped, Holistic Alignment Supercharges Diffusion Training

Ziqiao Wang[1][*]    Wangbo Zhao[1][*]    Yuhao Zhou[1]    Zekai Li[1]    Zhiyuan Liang[1]
Mingjia Shi[1]    Xuanlei Zhao[1]    Pengfei Zhou[1]    Kaipeng Zhang[2][†]
Zhangyang Wang[3]    Kai Wang[1][†]    Yang You[1]
[1]NUS HPC-AI Lab, [2]Shanghai AI Laboratory, [3]UT Austin

## Abstract

Diffusion Transformers (DiTs) deliver state-of-the-art image quality, yet their training remains notoriously slow. A recent remedy—*representation alignment* (REPA) that matches DiT hidden features to those of a *non-generative* teacher (e.g., DINO)—dramatically accelerates the *early* epochs but plateaus or even degrades performance later. We trace this failure to the **capacity mismatch**: once the generative student begins modeling the *joint* data distribution, the teacher's lower-dimensional embeddings and attention patterns become a straitjacket rather than a guide. We then introduce **HASTE** (**H**olistic **A**lignment with **S**tage-wise **T**ermination for **E**fficient training), a two-phase schedule that keeps the help and drops the hindrance. Phase *I* applies a *holistic* alignment loss that simultaneously distills *attention maps* (relational priors) and *feature projections* (semantic anchors) from the teacher into mid-level layers of the DiT, yielding rapid convergence. Phase *II* then performs one-shot termination that deactivates the alignment loss, once a simple trigger such as a fixed iteration is hit, freeing the DiT to focus on denoising and exploit its generative capacity. HASTE speeds up training of diverse DiTs without architecture changes. On ImageNet $256\times256$, it reaches the vanilla SiT-XL/2 baseline FID in **50 epochs** and matches REPA's best FID in **500 epochs**, amounting to a $28\times$ reduction in optimization steps. HASTE also improves text-to-image DiTs on MS-COCO, proving to be a simple yet principled recipe for efficient diffusion training across various tasks. Our code is available here.

## 1   Introduction

Diffusion Transformers (DiTs) are stunningly good—and stunningly slow. Recent variants such as DiT [37] and SiT [34] achieve state-of-the-art visual fidelity across a growing list of generative tasks [8, 29, 2, 9]. Unfortunately, their training incurs vast compute and wall-clock budgets because each update must back-propagate through hundreds of noisy denoising steps. A first wave of accelerators tackles this either by *architectural surgery*—linearized attention, masking or gating [56, 53, 12, 54, 26]—or by *training heuristics*, e.g. importance re-weighting of timesteps [49]. These interventions help, but often at the cost of specialized kernels or fragile hyper-parameter tuning.

**Representation alignment: early rocket, late parachute?** Recent work has demonstrated the effectiveness of leveraging external representations to accelerate diffusion model training—completely sidestepping the need for architectural modifications [55, 53, 45, 30]. A representative method, *Representation Alignment* (REPA) [55], projects an intermediate DiT feature map onto the embedding

---

[*]equal contribution (ziqiaow@u.nus.edu). Ziqiao, Wangbo, Zhangyang, and Kai are core contributors.
[†]corresponding author.

space of a powerful **non-generative vision encoder** such as DINOv2 [36], enforcing a cosine-similarity loss that bootstraps useful semantics during training. The gain is immediate: the student DiT latches onto global object structure and converges several times faster than a vanilla run. Yet REPA's help is not unconditional. Figure 1 removes the alignment loss after either 100K or 400K iterations. Stopping *late* (400K) *improves* FID over the always-on baseline; stopping *early* (100K) hurts—evidence that *REPA works until it doesn't*. Why?

**Our Conjecture: Capacity mismatch incurs the hidden turning point.** Diffusion models eventually model the *joint* data distribution, a harder objective than the *marginal/conditional* targets implicit in a frozen, non-generative encoder. Consequently, once the student has burned in, its own capacity overtakes the teacher's. Our gradient-angle analysis (Section 2.2) shows alignment and denoising objectives start *aligned* (acute angles), drift to orthogonality, then turn obtuse—signalling that continued alignment may become a harmful constraint.

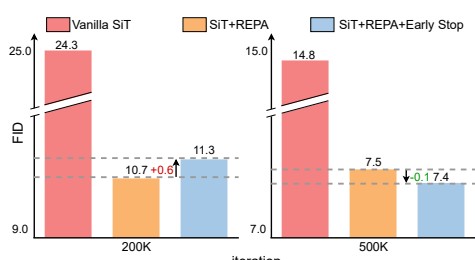

Figure 1: Training SiT-XL/2 on ImageNet 256×256. Adding REPA slashes FID early on, but its benefit fades and ultimately reverses; dropping the alignment loss mid-training restores progress.

**Simple Remedy: Holistic alignment, then release.** Two observations motivate our remedy. First, the teacher's *attention maps* encode relational priors that are as valuable as its embeddings [31, 20]; guiding only features leaves this structural knowledge untapped. Second, the alignment needs a *stage-wise schedule*: thick guidance early, zero guidance once gradients diverge. We therefore introduce **HASTE** (**H**olistic **A**lignment with **S**tage-wise **T**ermination for **E**fficient training). During Phase I we distill *both* projected features *and* mid-layer attention maps from DINOv2 into the DiT, giving the student relational and semantic shortcuts. Once a simple trigger (*e.g.*, fixed iteration) is hit, we enter Phase II: the alignment loss is disabled and training proceeds with the vanilla denoising objective. The recipe is two lines of code, no kernel changes.

**Contribution Summary.** Our findings refine the community's understanding of external representation guidance: it is immensely helpful early, but *must be let go* for the generative model to focus on specific tasks. We outline our contributions as follows.

- **Diagnosis.** We identify a capacity mismatch that flips REPA from accelerator to brake and quantify it via gradient-direction similarity.

- **Method.** We propose *holistic* (attention + feature) alignment combined with a *stage-wise termination* switch that deactivates alignment when it starts to impede learning.

- **Results.** On ImageNet 256×256 our schedule matches vanilla SiT-XL/2 in **50 epochs**, amounting to a 28× speed-up, and reaches REPA's best score in **500 epochs**. Gains replicate on COCO text-to-image generation task.

## 2   Method

Our framework, HASTE, couples two ingredients (see Figure 2): (i) **Holistic alignment**: a *dual-channel* distillation that supervises both projected features and attention maps; (ii) **Stage-wise termination**: a *single switch* that turns the alignment loss off once it ceases to help. We first recap REPA and attention alignment, then describe how we marry them and when we shut them off.

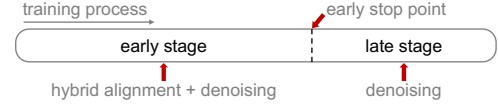

Figure 2: Overview of our framework. *Phase I* (left) distills *both* feature embeddings and attention maps from a frozen, non-generative teacher (DINOv2) into mid-level layers of the student DiT. When a simple trigger $\tau$ fires, the alignment loss is *disabled*; *Phase II* (right) then continues training with pure denoising.

## 2.1 Preliminaries

**Notation.** Let $\mathbf{x}$ be a clean image, $\tilde{\mathbf{x}}_t$ its noised version at timestep $t$, and $\mathbf{h}_t$ the hidden state of a Diffusion Transformer $\mathcal{G}_\theta$. A frozen, non-generative vision encoder $\mathcal{E}$ (DINOv2) produces patch embeddings $\mathbf{y} = \mathcal{E}(\mathbf{x})$ and self-attention matrices $\mathbf{A}^E$.

**Representation alignment (REPA).** A small MLP $g_\phi$ projects $\mathbf{h}_t$ into the encoder space. REPA [55] then aligns the projected state $g_\phi(h_t)$ with $y$ by maximizing token-wise cosine similarities:

$$\mathcal{L}_{\text{REPA}}(\theta, \phi) = -\mathbb{E}_{\mathbf{x}, \epsilon, t} \left[ \frac{1}{N} \sum_{n=1}^{N} \text{sim} \left( \mathbf{y}^{[n]}, g_\phi \left( \mathbf{h}_t^{[n]} \right) \right) \right] \tag{1}$$

This regularization is jointly optimized with the original denoising objective, to guide the more efficient training of diffusion transformers.

**Attention alignment (ATTA).** ATTA aims to transfer attention patterns from a pre-trained teacher model to a student model to guide the latter's training process [31]. For selected layers/heads $(i, j)$ we minimize token-wise cross-entropy between teacher and student attention.

## 2.2 Early Stop of Representation Alignment

**Gradient–based autopsy reveals state evolution.** Figure 1 already hinted that REPA's benefit peaks early and tapers off. To pinpoint *when* the auxiliary loss flips from help to hindrance, we inspect the *cosine similarity*

$$\rho_t = \cos\left(\nabla_\theta \mathcal{L}_{\text{diff}}, \nabla_\theta \mathcal{L}_{\text{REPA}}\right) \quad \in [-1, 1],$$

computed on the 8th block of SiT–XL/2 (the alignment depth used by REPA) over 960 ImageNet images (see details in Appendix A.1). A positive $\rho_t$ means the teacher pushes the student in roughly the *same* direction as denoising; negative one means the two losses actively fight.

Taking $t \leq 0.1$ for example, Figure 3 shows three distinct regimes:

1. *Ignition* (0–200 K iterations): $\rho_t$ starts with a relatively high level — REPA **adds** power; diffusion transformer profits from the teacher's guidance on representation learning.

2. *Plateau* (200 K–400 K iterations): $\rho_t$ decreases to nearly orthogonal level — objectives decouple; further REPA updates neither help nor hurt.

3. *Conflict* ($>$400 K iterations): $\rho_t$ exhibits negative values — gradients oppose; REPA now **erases** detail the student tries to learn.

The cross–over coincides with the iteration where Figure 1 shows FID curves diverging, confirming that gradient geometry is a faithful early-warning signal.

Figure 3: Cosine similarity between REPA and denoising gradients. Acute → orthogonal → obtuse: the auxiliary signal turns from booster to brake.

**Why does conflict arise? Capacity–mismatch view.** Once the student starts modeling the *joint* data distribution, it seeks high-frequency detail absent from the teacher's embeddings. A frozen encoder trained for invariant recognition discards such minutiae by design; forcing the student back into that lower-dimensional manifold yields destructive gradients. We see the same mismatch at the level of *diffusion timesteps.*

Figure 4 plots $\rho_t$ versus the diffusion time index. For mid-noise steps (e.g., $t = 0.5$) where the image is still blurry, gradients align. For late steps ($t \leq 0.1$)—responsible for textures and fine grain [23]—they are near-orthogonal *from the start*. This indicates that teacher guidance is intrinsically global; when the denoiser must polish pixels, the encoder has little to teach.

Figure 4: Gradient similarity as function of diffusion timestep $t$. At $t = 0.1$ (high-detail phase) the two losses already conflict even early in training.

We sharpen this claim by feeding the teacher *low-frequency only* versions of each image (Figure 5). Early FID improves almost identically to vanilla REPA, proving that the speed-up stems from *coarse semantic scaffolding*; high-frequency cues are less relevant to REPA's benefit.

**Take-away.** REPA supplies valuable *global* context but obstructs *local* detail once the student matures. Hence, alignment should be **transient** for further improvement.

**Fix: Stage-wise termination.** Let $\tau$ denote the termination iteration around which $\rho_t$ exhibits low similarity and the alignment provides limited benefit. We then *discard* the auxiliary alignment loss:

$$\mathcal{L}(\theta, \phi) = \begin{cases} \mathcal{L}_{\text{diff}} + \mathcal{L}_R, & n < \tau, \\ \mathcal{L}_{\text{diff}}, & n \geq \tau, \end{cases} \quad (2)$$

where $\mathcal{L}_R$ may itself be the holistic combo of feature (Section 2.1) and attention (Section 2.3) losses. A fixed $\tau$ works reliably well, but the gradient rule adds robustness (Appendix A.2).

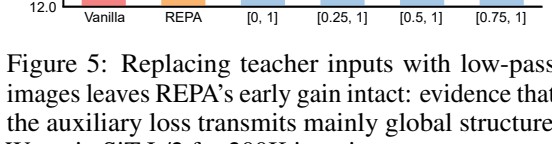

Figure 5: Replacing teacher inputs with low-pass images leaves REPA's early gain intact: evidence that the auxiliary loss transmits mainly global structure. We train SiT-L/2 for 200K iterations.

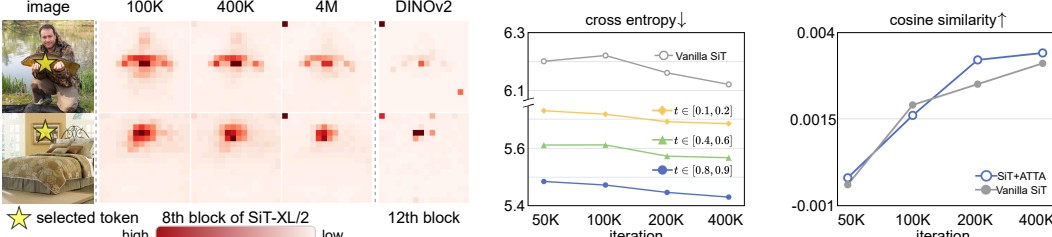

(a) Visualization of attention maps from DINOv2-B and SiT-XL/2+REPA at different training iterations. (b) Attention alignment progress with REPA alone. (c) Feature alignment progress with ATTA alone.

Figure 6: Evaluating cross-effects between feature and attention alignment. (a) Attention map visualization of selected tokens for SiT-XL/2+REPA and DINOv2-B. (b) Alignment depth at 5, we track attention map cross-entropy between the 12th-layer of DINOv2-B and the 5th-layer of SiT-B/2. (c) Attention maps from 3rd–5th layers of SiT-B/2 are aligned with those from 8th, 10th, and 12th layer of DINOv2-B. Since ATTA alone does not optimize the projector, we directly compute cosine similarity between the DINOv2-B features and the 5th-layer hidden states of SiT (without projection).

## 2.3 Holistic Alignment by Integration Attention

**Rationale: Why attending to *attention*?** Compared with *token embeddings*, self–attention matrices reveal *where* a transformer routes information at each layer—its "inference pathways" in the sense of Hoang et al. [20]. These pathways encode rich relational priors: object–part grouping, long–range symmetry, and background–foreground segregation emerge as distinct heads in DINOv2, even though the model was trained without labels. Critically, such routing information is *orthogonal* to the static content captured by features: two models can share identical patch embeddings but attend to them in entirely different patterns, leading to divergent downstream behavior.

Recent evidence echoes: Li et al. [31] show that distilling *only* attention maps from a high-capacity teacher to a randomly initialized ViT is more effective than transferring *only* embeddings, in recovering the teacher's linear probe precision on ImageNet. The asymmetry suggests that attention acts as a **structural prior**: once the model is taught *how to look*, it can relearn *what to look at* rapidly.

For diffusion transformers, they must integrate global spatial cues (layout, object boundaries) across hundreds of tokens for effective representation construction. While feature alignment (REPA) accelerates the learning process by injecting semantic anchors, the structural knowledge remains underexploited. Attention alignment targets the complementary regime: it transfers the *global routing template* to DiT, thereby enabling precise spatial and global information guidance.

**Motivational experiments.** To disentangle the respective contributions of *features* and *attention*, we probe the two signals in isolation:

- *(i)* **Feature alignment only (REPA).** Figure 6a shows that REPA gradually makes SiT heads resemble those of the teacher. However, the convergence of attention patterns is slow and incomplete (see cross-entropy trend in Figure 6b).

- *(ii)* **Attention alignment only (ATTA).** Aligning attention maps alone can also pull the student's hidden features toward the teacher's embedding space (Figure 6c) and yields a training-speed boost on par with REPA (see details in Section 3.4).

**Takeaway.** REPA bootstraps *semantics* but leaves routing under-constrained; ATTA nails routing but still requires the conditional gates to be learned from scratch. Their complementary effects motivate combining both. For a chosen set $\mathcal{S}$ of student–teacher layer pairs $(\ell_s, \ell_t)$ and the $M$ heads,

$$\mathcal{L}_{\text{ATTA}} \;=\; \frac{1}{|\mathcal{S}|\,M} \sum_{(\ell_s,\ell_t)\in\mathcal{S}} \sum_{m=1}^{M} \mathcal{H}\!\Big(\text{softmax}\big(Q_s^{\ell_s,m} K_s^{\ell_s,m\top}\big), \text{softmax}\big(Q_t^{\ell_t,m} K_t^{\ell_t,m\top}\big)\Big), \quad (3)$$

where $\mathcal{H}$ is token–wise cross-entropy.

**Where and when to align attention?** We distill teacher heads *only* into **intermediate** student blocks (e.g., SiT-XL/2 blocks 4–7). Two empirical observations justify this selective schedule:

- *(i)* **Shallow mismatch.** Early DiT layers ingest *Gaussian-noisy latents*; their representations are dominated by variance normalization and channel folding rather than semantics. Supervising those layers with *pixel-space* attention from a clean-image encoder is therefore off-manifold. In practice, forcing attention on too many shallow layers destabilizes the loss and raises FID.

- *(ii)* **Deep freedom.** The ultimate objective of DiT is denoising for high-quality generation, rather than representation learning. The last blocks are responsible for translating high-level structure into precise generation update. Thus, these blocks should remain dedicated to the denoising objective, unregularized.

Aligning mid-layers strikes the sweet spot: they are late enough that latents carry discernible semantics, yet early enough that constraining their routing gives downstream blocks a clean, well-organized feature tensor to refine.

## 2.4 Final Recipe: HASTE

**Where we align.** *Attention maps* from the teacher are distilled into a *range* of mid-depth DiT blocks; *features* follow the original REPA setting—one projection at a single mid-layer. Neither the shallow noise processing blocks nor the final denoising blocks are regularized.

**What we align.** During Phase I (iterations $n < \tau$) we apply a *hybrid* auxiliary loss. $\lambda_R$ and $\lambda_A$ are weight coefficients for balancing two regularizations.

$$\mathcal{L}_R = \lambda_R \, \mathcal{L}_{\text{REPA}} + \lambda_A \, \mathcal{L}_{\text{ATTA}}. \quad (4)$$

**When we stop.** At the single switch point $\tau$—chosen as a fixed iteration or the gradient-angle trigger from §2.2—*both* terms in (4) are dropped and training proceeds with the vanilla denoising objective.

This three-line schedule constitutes *HASTE*: it retains REPA's semantic anchoring, adds ATTA's routing prior, and removes all auxiliary constraints once they turn counter-productive.

# 3 Experiments

## 3.1 Setup

**Models and datasets.** Following REPA [55], we conduct experiments on three diffusion transformers: SiT [34], DiT [37], and MM-DiT [8]. ImageNet [4] and MS-COCO 2014 [32] datasets are used for class-to-image and text-to-image generation tasks, respectively. Moreover, we employ a pre-trained DINOv2-B [36] as the representation model to extract high-quality features and attention patterns.

**Implementation details.** We use a training batch size of 256 and SD-VAE [40] for latent diffusion, and set $\lambda_R = 0.5$ following REPA to ensure a fair comparison. Additionally, we also adopt the SDE Euler-Maruyama sampler with NFEs = 250 for image generation on SiT and DiT. We set $\lambda_A = 0.5$ as the weight of attention alignment. We use NVIDIA A100 and H100 compute workers.

**Evaluation metrics.** For ImageNet experiments, we sample 50K images to assess the performance, leveraging evaluation protocols provided by ADM [5] to measure FID [16], sFID [35], IS [42], and Precision and Recall [27]. For text-to-image generation, we follow the settings defined in [1].

## 3.2 Experiments on ImageNet 256 × 256

**Setting.** In this experiment, we set the termination point $\tau = 100K$ iteration (around 20 epochs) for SiT-B/2 and $\tau = 250K$ iteration (around 50 epochs) for large and xlarge size models. while all other settings remain at their default values.

**Results without classifier-free guidance.** As shown in Table 1, HASTE demonstrates significant acceleration performance, consistently outperforming REPA on both SiT-XL and DiT-XL. This validates the superiority of stage-wise termination and holistic alignment. Notably, on SiT-XL, HASTE achieves an FID of 8.39 with only 250K iterations (50 epochs), matching the performance of vanilla SiT-XL with 1400 epochs, representing a 28× acceleration. Similarly, on DiT-XL, our approach surpasses the original DiT-XL trained with 1400 epochs, using only 80 epochs.

**Results with classifier-free guidance.** We also evaluate the generation performance of SiT-XL+HASTE at different epochs with classifier-free guidance (CFG) [17] applying guidance interval [28]. As shown in Table 1, HASTE outperforms most of the baselines in only 400 epochs, and can achieve a comparable FID score to REPA with 500 epochs, which proves that in later training stages, the denoising objective itself is also able to lead diffusion transformers to satisfactory generation capability.

**Qualitatively comparison.** We also provide representative visualization results from SiT-XL/2 with REPA and HASTE in Figure 8, respectively. Our method achieves better semantic information and detail generation at early training stages.

| method | epoch | FID↓ | sFID↓ | IS↑ | Prec.↑ | Rec.↑ |
|---|---|---|---|---|---|---|
| *Without* Classifier-free Guidance (CFG) | | | | | | |
| MaskDiT | 1600 | 5.69 | 10.34 | **177.9** | **0.74** | 0.60 |
| DiT | 1400 | 9.62 | 6.85 | 121.5 | 0.67 | 0.67 |
| SiT | 1400 | 8.61 | 6.32 | 131.7 | 0.68 | 0.67 |
| DiT+REPA | 170 | 9.60 | - | - | - | - |
| SiT+REPA | 800 | 5.90 | 5.73 | 157.8 | 0.70 | **0.69** |
| FasterDiT | 400 | 7.91 | 5.45 | 131.3 | 0.67 | **0.69** |
| MDT | 1300 | 6.23 | 5.23 | 143.0 | 0.71 | 0.65 |
| DiT+**HASTE** | 80 | 9.33 | 5.74 | 114.3 | 0.69 | 0.64 |
| SiT+**HASTE** | 50 | 8.39 | 4.90 | 119.6 | 0.70 | 0.65 |
|  | 100 | **5.31** | **4.72** | 148.5 | 0.73 | 0.65 |
| *With* Classifier-free Guidance (CFG) | | | | | | |
| MaskDiT | 1600 | 2.28 | 5.67 | 276.6 | 0.80 | 0.61 |
| DiT | 1400 | 2.27 | 4.60 | 278.2 | **0.83** | 0.51 |
| SiT | 1400 | 2.06 | 4.50 | 270.3 | 0.82 | 0.59 |
| FasterDiT | 400 | 2.03 | 4.63 | 264.0 | 0.81 | 0.60 |
| MDT | 1300 | 1.79 | 4.57 | 283.0 | 0.81 | 0.61 |
| DiT+TREAD | 740 | 1.69 | 4.73 | 292.7 | 0.81 | 0.63 |
| MDTv2 | 1080 | 1.58 | 4.52 | **314.7** | 0.79 | **0.65** |
| SiT+REPA | 800 | 1.42 | 4.70 | 305.7 | 0.80 | **0.65** |
| SiT+**HASTE** | 100 | 1.74 | 4.74 | 268.7 | 0.80 | 0.62 |
|  | 400 | 1.44 | 4.55 | 293.4 | 0.80 | 0.64 |
|  | 500 | 1.42 | **4.49** | 299.5 | 0.80 | **0.65** |
|  | 600 | **1.41** | 4.51 | 296.9 | 0.80 | **0.65** |

Table 1: System-level comparison on ImageNet 256 × 256. ↑ and ↓ denote higher and lower values are better, respectively. **Bold font** denotes the best performance.

## 3.3 Text-to-Image Generation Experiment

**Setting.** To validate our approach in text-to-image generation tasks, we apply HASTE to MM-DiT [8], a widely used architecture, and train it on the MS-COCO 2014 dataset [32] following REPA. In practice, we set termination point $\tau = 200K$ for HASTE. Moreover, we only perform attention alignment with the $QK^T$ matrix generated from input image to avoid affecting the textual process.

**Quantitative results.** In Table 2, we compare our method with the original MM-DiT and MM-DiT+REPA using SDE sampler with NFEs = 250. Results reflect that HASTE consistently out-

performs its counterparts and alignment termination leads to better performance, validating the generalizability of our holistic alignment and termination strategy in text-to-image generation tasks.

## 3.4 Ablation Studies

In this section, we conduct extensive experiments and comparisons across different SiT models on ImageNet 256×256, to further support our analysis and claims in Section 2. We consistently use the SDE Euler-Maruyama sampler (NFEs = 250) without classifier-free guidance.

**Effectiveness of ATTA and termination.** To validate the effectiveness of termination and Attention Alignment, we evaluate the performance of SiT-XL/2 with different methods applied before and after the termination point (50 epoch) and present the results in Table 3. Firstly, at both 40 and 100 epochs, we observe that using only Attention Alignment can also obtain a similar acceleration to REPA. Moreover, the holistic alignment leads to better performance at 40 epoch, which is consistent with our hypothesis in Section 2.3 that the two methods have complementary potentials.

| model | iteration | term. | FID↓ |
|---|---|---|---|
| MM-DiT | 150K | - | 5.26 |
| +REPA | 150K | - | **4.16** |
| +REPA | 250K | - | 4.28 |
| **+HASTE** | 150K | - | 4.09 |
| **+HASTE** | 250K | - | 4.10 |
| **+HASTE** | 250K | 200K | **4.06** |

Table 2: FID↓ results of text-to-image generation on MS-COCO. HASTE consistently outperforms REPA to accelerate the training of MM-DiT.

However, the acceleration of such integration gets inferior to REPA alone at 100 epoch. We assume that consistently applying holistic alignment leads to over-regularization in later training stages. And the performance gets improved eventually with the termination strategy applied at 50 epoch.

**Different termination iterations $\tau$.** In this section, we analyze the impact of $\tau$ across varying model sizes. First, we conduct experiments in Table 4 to further explore the effect of termination. The results reflect that stage-wise termination also leads to better generation quality on SiT-B/2 and SiT-L/2. For SiT-XL/2, interestingly, while $\tau = 400$ K demonstrates a lower FID at 400K iteration, $\tau = 250$ K model ultimately delivers superior performance when evaluated at 500K iteration.

As shown in Table 3 and Table 4, although holistic alignment achieves better performance at 400K iteration, consistently regularizing the model leads to reduced performance. While termination at $\tau = 400$ K alleviates such a trend, its performance at 500K iteration is still inferior to that of $\tau = 250$ K. Therefore, we hypothesize that the acceleration effect gradually diminishes before 400K iteration, and the stage-wise termination, such as at $\tau = 250$ K, can help to alleviate the over-regularization.

| epoch | REPA | ATTA | term. | FID↓ | sFID↓ | IS↑ |
|---|---|---|---|---|---|---|
| 40 | × | × | × | 24.3 | 5.08 | 56.1 |
|  | ○ | × | × | 10.7 | **5.02** | 103.9 |
| 40 | × | ○ | × | 13.6 | **5.02** | 89.7 |
|  | ○ | ○ | × | **9.9** | 5.04 | **108.8** |
| 100 | × | × | × | 14.8 | 5.18 | 84.9 |
|  | ○ | × | × | 7.5 | 5.11 | 130.1 |
| 100 | × | ○ | × | 8.5 | 5.00 | 120.7 |
|  | ○ | ○ | × | 8.1 | 5.20 | 126.1 |
|  | ○ | ○ | ○ | **5.3** | **4.72** | **148.5** |

| model | iteration | $\tau$ | FID↓ | sFID↓ | IS↑ |
|---|---|---|---|---|---|
| SiT-B/2 | 400K | - | 21.3 | 6.80 | 69.9 |
| **+HASTE** |  | 100K | **19.6** | **6.38** | **73.0** |
| SiT-L/2 | 400K | - | 8.9 | 5.18 | 119.0 |
| **+HASTE** |  | 250K | **7.9** | **5.08** | **124.8** |
| SiT-XL/2 | 400K | - | **5.5** | **4.74** | **144.4** |
| **+HASTE** |  | 250K | 7.3 | 5.05 | 128.7 |
| SiT-XL/2 | 500K | - | 8.1 | 5.20 | 126.1 |
| **+HASTE** |  | 250K | **5.3** | **4.72** | **148.5** |
|  |  | 400K | 7.4 | 5.10 | 128.8 |

Table 3: Comparison of different methods applied to SiT-XL/2. ○ and × denote methods applied or not, respectively. Results reflect that our termination and holistic alignment strategies are effective.

Table 4: Comparison of applying termination or not across different model sizes of SiT. $\tau$ denotes termination point. We find the termination strategy contributes to better performance eventually.

Taking SiT-XL/2 for example, we carefully assess the effect of different $\tau$. We observe performance progresses slowly after 250K iteration (see Figure 7a). And the gradient cosine similarity between holistic alignment and denoising has shown negative values at late diffusion timesteps (see details in

Appendix A.1). Consequently, we consider termination near this threshold: results at 400K iteration in Figure 7b indicate that early stopping at $\tau = 250K$ yields better performance.

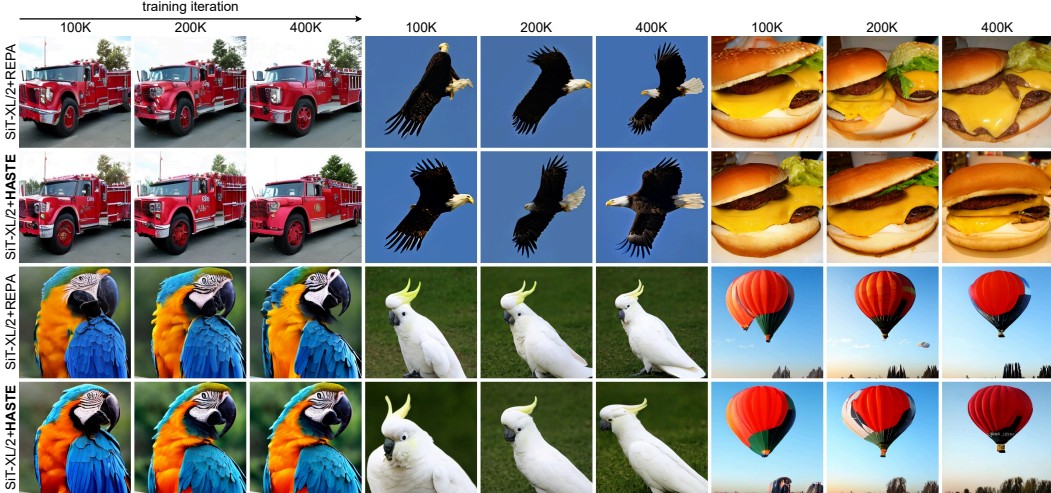

Figure 8: HASTE improves visual scaling. We compare images generated by SiT-XL/2+REPA and SiT-XL/2+HASTE (ours) at different training iterations. For both models, we use the same seed, noise, and sampling method with a classifier-free guidance scale of 4.0.

**Different Attention Alignment loss weight** $\lambda_A$**.** We evaluate the sensitivity of model to the attention alignment loss weight $\lambda_A$ in Equation 4 with SiT-L/2 as an example.

As shown in Table 5, HASTE consistently improves the performance of SiT-L/2 at 400K iteration across different values of $\lambda_A$, indicating that attention alignment provides relatively stable benefits. We note that larger weights can lead to reduced performance. Therefore, we choose $\lambda_A = 0.5$ in our primary experiments.

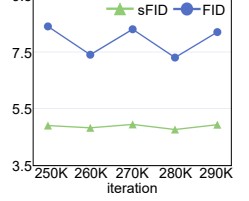
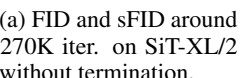
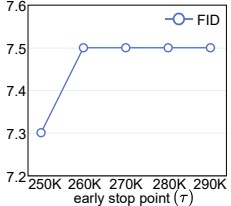

(a) FID and sFID around 270K iter. on SiT-XL/2 without termination.

(b) FID at 400K iter. with termination point $\tau$ around 270K.

Figure 7: Comparison of different termination point $\tau$ on SiT-XL/2. We observe the training oscillation after 250K iteration. Using $\tau = 250$ K leads to better performance at 400K iteration.

**Selection of alignment layers**. We try different transfer layers for HASTE on SiT-L/2 in Table 6. For brevity, we denote layers from SiT and DINO as layer-S and layer-D, respectively. Additionally, we use $[\cdot]_S$ and $[\cdot]_D$ to specify particular layer indices (counting from 0).

Firstly, we find that enough deeper layers should get involved for optimal performance. As shown in Table 6, when choosing only two layers of each model for alignment, namely $[10, 11]_D$ and $[6, 7]_S$, the performance is inferior to choosing four layers. Additionally, results reflect that enough shallow layers should be left for processing the noisy inputs in the latent space. We can observe that the distillation of $[8, 9, 10, 11]_D$ to $[4, 5, 6, 7]_S$ achieves a better FID without including $[6, 7]_D$ to $[2, 3]_S$.

| model | $\lambda_A$ | FID↓ | sFID↓ | IS↑ |
|---|---|---|---|---|
| | 0.5 | **7.9** | **5.08** | **124.8** |
| SiT-L/2 | 1.0 | 8.6 | 5.29 | 120.0 |
| +**HASTE** | 1.5 | 8.7 | 5.23 | 119.3 |
| | 3.0 | 9.0 | 5.34 | 116.9 |

Table 5: ATTA weight $\lambda_A = 0.5$ leads to better performance on SiT-L/2 at 400K iteration.

| model | layer-D | layer-S | FID↓ | sFID↓ | IS↑ |
|---|---|---|---|---|---|
| | $[10, 11]_D$ | $[6, 7]_S$ | 8.9 | 5.31 | 119.3 |
| SiT-L/2 +**HASTE** | $[8, 9, 10, 11]_D$ | $[4, 5, 6, 7]_S$ | **7.9** | **5.08** | **124.8** |
| | $[6, 7, 8, 9, 10, 11]_D$ | $[2, 3, 4, 5, 6, 7]_S$ | 8.3 | 5.12 | 121.3 |

Table 6: Comparison of HASTE with different choices of layers on SiT-L/2 at 400K iteration. While transferring attention maps for more deep layers provides greater benefits, we need to preserve enough shallow layers to process latent input.

Our findings align with the observations of attention transfer on ViTs reported in [31]: transferring more attention maps from deeper layers provides greater benefits, and ViTs can learn low-level features well when guided on how to integrate these features into higher-level ones.

## 4 Related Work

### 4.1 Accelerating Training Diffusion Transformers

To accelerate the training of diffusion transformers, existing methods can be broadly classified into two categories: architectural modifications and representation enhancements.

**Architecture modification.** These methods focus on directly improving the efficiency of the model architecture. For example, SANA series [51, 52], DiG [58], and LiT [47] introduce Linear Attention [22, 50, 3] to improve the efficiency of diffusion transformers. Additionally, methods like MaskDiT [56] and MDT [10, 11] introduce masked image modeling [14] to reduce the cost during training.

**Representation incorporation.** In contrast to architecture modifications, these methods do not require designing specialized structures and instead leverage external representations to achieve acceleration. For instance, REPA [55] observes the difficulty in learning effective representations for diffusion models [43, 18, 44], which hinders the training efficiency. To address this, REPA proposes to align the internal features of diffusion transformers with the output of pre-trained representation models, and significantly accelerates the training process.

Furthermore, recent works [53, 45, 30] have also achieved better results based on representation methods. For example, U-REPA [45] improves REPA with a manifold alignment loss, and demonstrates its effectiveness on U-Nets [41]. External representations can also help enhance generation and reconstruction capabilities of VAE, such as in VA-VAE [53] and E2E-VAE [30].

Unlike these methods, our research focuses mainly on the diffusion transformer itself. We investigate the relationship between external representation guidance and the self-improvement of diffusion transformers, and propose to remove the regularization at an appropriate training stage.

### 4.2 Attention Transfer for Vision Transformers

The attention mechanism [46] has been shown to provide vision models, such as Vision Transformers (ViTs) [6], with strong adaptability and scalability across various tasks. While prior works [15, 13] have achieved improved downstream performance by leveraging entire pre-trained models, Li et al. [31] demonstrates that the attention patterns learned during pre-training are sufficient for ViTs to learn high-quality representations from scratch, achieving performance comparable to fine-tuned models on downstream tasks. Consequently, attention distillation [31, 48, 39] has been proposed to transfer knowledge efficiently.

The transfer of attention maps has been extensively studied in Vision Transformers (ViTs), but remains underexplored in diffusion transformers. While recent work [57] applies attention distillation for characteristics transfer tasks using diffusion models, its explorations remain in the sampling process. Moreover, the relationship between attention mechanisms in ViTs and diffusion transformers requires further investigation. In this work, we demonstrate that attention maps from a pre-trained ViT can effectively guide the learning process of diffusion transformers.

## 5 Conclusion

In this paper, we have proposed HASTE, a simple but effective way to improve the training efficiency of diffusion transformers. Specifically, we identify the capacity mismatch and reveal that *representation alignment is not always beneficial throughout the training process*. In addition, we analyze the stages when feature alignment is most effective and investigate the dilemma between external feature guidance and internal self-improvement of diffusion transformers. We prove that HASTE can significantly accelerate the training process of mainstream diffusion transformers, such as SiT and DiT. We hope our work will further reduce the cost for researchers to train diffusion transformers, and broaden the application of diffusion models in downstream tasks.

**Limitations and future work.** We mainly focus on diffusion transformers in latent space for image generation. Explorations of HASTE with pixel-level diffusion [5, 24], or in video generation tasks [19] would be exciting directions for future work. Additionally, HASTE may also be incorporated with other methods [53, 30] on different model architectures [45].

Faster generative model training could potentially lower barriers for harmful content generation. As a technical contribution focused on training efficiency, our work does not directly address broader AI safety challenges, which we acknowledge as important future work.

It is meaningful to further explore attention alignment over cross-attention maps between MM-DiT and pre-trained multi-modal encoders. The development of more robust and efficient *automatic* termination triggers—possibly guided by gradient-based signals—remains an open direction.

**Acknowledgement.** We sincerely appreciate Liang Zheng and Ziheng Qin for valuable discussions and feedback during this work. We acknowledge anonymous reviewers for insightful questions and discussions. Our work is sponsored by NUS startup grant (Presidential Young Professorship), Singapore MOE Tier-1 grant, ByteDance grant, NUS ARTIC grant, Apple grant, Alibaba grant, Google Research and Google grant for TPU usage.

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

# A    Additional Results

## A.1    Gradient Angle Diagnosis

We provide detailed results of cosine similarity between REPA [55] and denoising gradients. In Figure 9, we separately compute gradients of the feature alignment and the denoising objective for SiT-XL/2 [34] and compare the cosine similarity of their directions at different training iterations. Specifically, we randomly sample 960 images from the training dataset of ImageNet [4] for the comparison and take gradients of parameters in the eighth block of SiT-XL/2 for example (REPA sets the default alignment depth as 8).

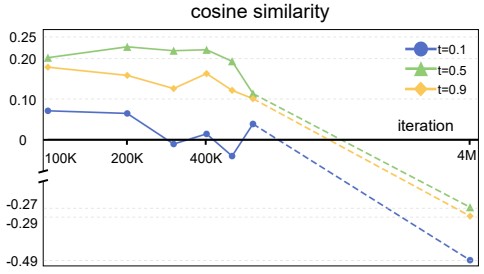

Figure 9: Gradient cosine similarity between REPA and the denoising objective.

| iteration | t = 0.02 | t = 0.04 | t = 0.06 | t = 0.08 | t = 0.10 |
|---|---|---|---|---|---|
| 100K | 0.0070 | 0.0064 | 0.0327 | 0.0525 | 0.0692 |
| 200K | 0.0350 | 0.0476 | 0.0434 | 0.0568 | 0.0628 |
| 300K | -0.0235 | -0.0324 | -0.0316 | -0.0044 | -0.0116 |
| 400K | -0.1236 | -0.1056 | -0.1133 | 0.0232 | 0.0130 |
| 500K | 0.0346 | -0.0368 | -0.0246 | -0.0063 | -0.0409 |
| 600K | -0.1185 | -0.0546 | 0.0645 | -0.0039 | 0.0372 |
| 4M | -0.2065 | -0.1279 | -0.1928 | -0.3621 | -0.4942 |

Table 7: Detailed cosine similarity results of the $8^{\text{th}}$ block in SiT-XL/2 at $t \leq 0.10$.

We first observe a relatively high cosine similarity, representing an acute angle between gradients of the two objectives. However, the similarity shows a decreasing trend as the training progresses, and the angle becomes nearly orthogonal at the intermediate stage (around 400K iteration). Furthermore, we find that the similarity becomes obviously negative at the final training stage, such as at 4M iteration, indicating that there might be some potential conflict between REPA and diffusion loss.

In addition to training iterations, we also find a feature alignment gap over different diffusion timesteps: As reported in [55], a well-trained DiT [37] or SiT exhibits a higher feature alignment at the intermediate diffusion timesteps, while the alignment is notably weaker at those closer to the data distribution, i.e., nearby the sampling results, such as $t = 0.1$ for SiT. We observe a similar trend in our gradient similarity comparison. According to diffusion sampling properties, the initial steps starting from noise mainly contribute to global fidelity, namely the basic outline of images, while the steps closer to the data are to refine microscopic details such as textures [23]. We hypothesize that the diffusion transformer eventually needs to refine its own representations for detail generation beyond learning directly from external features.

| iteration | t = 0.02 | t = 0.05 | t = 0.07 | t = 0.1 | t = 0.2 | t = 0.5 | t = 0.9 |
|---|---|---|---|---|---|---|---|
| 100K | -0.0138 | -0.0131 | -0.0068 | 0.0129 | 0.0488 | 0.0541 | -0.0093 |
| 200K | -0.0423 | -0.0674 | -0.0719 | -0.0491 | 0.0068 | 0.0801 | 0.0099 |
| 250K | -0.0323 | -0.0597 | -0.0598 | -0.0599 | -0.0264 | 0.0354 | 0.0419 |
| 260K | -0.0232 | -0.0331 | -0.0243 | -0.0034 | 0.0436 | 0.0729 | 0.0065 |
| 270K | 0.0029 | 0.0152 | 0.0113 | 0.0097 | 0.0419 | 0.0554 | 0.0233 |
| 280K | -0.0263 | -0.0131 | -0.0031 | 0.0011 | 0.0217 | 0.0455 | -0.0176 |
| 290K | -0.0524 | 0.0199 | 0.0308 | 0.0532 | 0.0832 | 0.0550 | 0.0111 |

Table 8: Detailed gradient cosine similarity results between holistic alignment and denoising objectives on the $8^{\text{th}}$ block of SiT-XL/2 at different training iterations.

For our method, HASTE, we also examine the gradient cosine similarity between holistic alignment and denoising. The similarity trend serves as a kind of reference for our termination strategy.

## A.2 Gradient Angle Trigger

In Section 2.2, we introduce the gradient-angle rule for robustness and theoretical insight of HASTE. In this section, we further explain the role of gradient signals, and then provide a plain example of HASTE with automatic gradient-angle trigger.

**Hybrid termination strategy.** In practice, we find setting a fixed termination iteration $\tau$ for HASTE actually performs reliably well in ImageNet and MS-COCO experiments. Gradient threshold serves as a supplemental alternative, adding adaptability to different situations. Moreover, gradient analysis helps us to locate the termination point. As we state in Section 3.4, we choose to stop around 250K iteration when we observe limited alignment benefits and relatively low gradient cosine similarities.

**Gradient-angle trigger (plain implementation).** We train SiT-B/2 on ImageNet $256 \times 256$ following the original settings of HASTE. We start evaluating the gradient cosine similarity between holistic alignment and denoising objectives every 10K iterations after 50K iteration, when the training process has been relatively stable. At each evaluation, we use selected timesteps $t$ at various noise levels with 2048 images randomly sampled from ImageNet, and average the similarities. When the average similarity falls below a threshold $\delta$, we terminate the holistic alignment. In the plain example, we perform gradient evaluation on the 5th SiT block (counting from 1). We select $t = 0.02$, $t = 0.1$, $t = 0.5$ and $t = 0.8$, and set $\delta = -0.05$. As shown in Tables 9 and 10, our automatic trigger terminates the holistic alignment at 80K iteration, and achieves better performance at 100K iteration.

**Limitations.** In practice, we find it challenging to design such trigger for precise termination because of: (i) **Additional overhead**: gradient evaluation introduces extra computational costs, limiting the practical evaluation frequency. (ii) **Hyperparameter setting**: the automatic trigger is sensitive to hyperparameters, such as block choice, timestep selection, and the threshold.

Therefore, since monitoring model performance during training is a standard practice in modern deep learning workflows, *it's more practical to integrate the periodic evaluation of both metrics and gradient cosine similarity to determine the optimal termination point*.

| timestep | iteration | | | |
| --- | --- | --- | --- | --- |
| | 50K | 60K | 70K | 80K |
| 0.02 | -0.0174 | 0.0418 | 0.0872 | -0.0032 |
| 0.1 | 0.0126 | -0.0767 | -0.0148 | -0.1155 |
| 0.5 | 0.0134 | 0.0106 | -0.0848 | -0.1272 |
| 0.8 | 0.0034 | 0.0406 | 0.1025 | -0.0212 |
| average | 0.0030 | 0.0041 | 0.0225 | **-0.0668** |

Table 9: Gradient cosine similarity between alignment and denoising objectives. **Bold font** denotes the average value below threshold. We remove alignment at 80K iteration.

| method | iteration | encoder | FID↓ | sFID↓ | IS↑ |
| --- | --- | --- | --- | --- | --- |
| SiT-B/2 | 100K | - | 63.46 | 7.31 | 20.6 |
| +REPA | 100K | DINOv2-B | 49.50 | **7.00** | 27.5 |
| +REPA | 100K | CLIP-ViT-B | 54.92 | 7.63 | 24.7 |
| **+HASTE** | 100K | DINOv2-B | **39.86** | 7.16 | **35.8** |
| **+HASTE** | 100K | CLIP-ViT-B | 48.74 | 7.92 | 28.7 |

Table 11: Results of holistic alignment on SiT-B/2 with CLIP-ViT-B/16 as external encoder. While HASTE with CLIP consistently accelerates training, using DINOv2-B achieves better performance.

| method | iteration | term. | FID↓ | sFID↓ | IS↑ |
| --- | --- | --- | --- | --- | --- |
| SiT-B/2 | 100K | - | 63.46 | 7.31 | 20.6 |
| **+HASTE** | 100K | - | 39.86 | 7.16 | 35.8 |
| **+HASTE** | 100K | 80K | **38.69** | **6.88** | **36.4** |

Table 10: Results of SiT-B/2+HASTE with gradient angle trigger. Alignment termination at 80K iteration leads to better performance.

| method | iteration | term. | FID↓ | sFID↓ | IS↑ |
| --- | --- | --- | --- | --- | --- |
| SiT-B/2 | 150K | - | 52.71 | 7.06 | 26.2 |
| **+HASTE** | 150K | - | 38.36 | 7.23 | 38.2 |
| **+HASTE** | 150K | 100K | **36.13** | **6.95** | **40.6** |

Table 12: Validation of termination strategy on HASTE with CLIP-ViT-B/16. Alignment termination at 100K iteration achieves better performance.

### A.3 Experiments with CLIP

In this section, we take CLIP-ViT-B/16 [38] as an example to investigate the impact of different pre-trained encoders on ImageNet $256 \times 256$. Additionally, we propose a solution for performing alignment when there is a mismatch in patch numbers between SiT and the external encoder.

**Setting.** We use SiT-B/2 for fast validation following the original settings of HASTE. Since SiT-B/2 and CLIP-ViT-B/16 possess different patch numbers (i.e., $16 \times 16$ and $14 \times 14$), we perform additional spatial interpolations to align the dimensions:

- *(i)* **Feature map interpolation.** Given SiT-B/2 block hidden states of shape $[16 \times 16, 768]$, instead of directly merging channels from 256 into 196, we first reshape the sequence to $[16, 16, 768]$ to preserve the 2D spatial topology of image patches. Subsequently, we apply *bilinear interpolation* to downsample the feature from spatial dimensions $[16, 16]$ to $[14, 14]$.
- *(ii)* **Attention map interpolation.** Similarly, given SiT-B/2 block attention map of shape $[h, 16 \times 16, 16 \times 16]$, we reshape it into a spatial tensor $[h, 16, 16, 16, 16]$, where the dimensions represent head number, query height, query width, key height and key width respectively. Subsequently, we sequentially apply *bilinear interpolation* to key and query spaces to downsample the attention map from dimensions $[16, 16, 16, 16]$ to $[14, 14, 14, 14]$.

**Results (w/o cfg).** As shown in Tables 11 and 12, our implementations of both REPA and HASTE with CLIP-ViT-B/16 can accelerate the training process of SiT-B/2. However, using DINOv2-B achieves better acceleration performance, which aligns with the findings in [55]: *when the diffusion transformer is aligned with a pre-trained encoder that offers more semantically meaningful representations, the model exhibits enhanced generation performance*. At 150K iteration, the holistic alignment early-stopped at 100K iteration achieves better performance, validating our termination strategy.

Consequently, we validate the generalizability of HASTE over different pre-trained encoders. Moreover, apart from directly varying the size of input image to the external encoder, we provide an interpolation solution to address the patch number mismatch between SiT and the encoder.

### A.4 ImageNet $512 \times 512$ Experiment

We conduct an additional experiment on ImageNet $512 \times 512$ to validate the scalability of HASTE to image resolution. We follow the original settings of HASTE except $\tau = 300K$ iteration. Specifically, we resize the input image to DINOv2 from $512 \times 512$ to $448 \times 448$ resolution following REPA. In practice, we use stable diffusion VAE [40] to compress the input image to SiT during training process, instead of pre-computing the latent vectors due to storage limitations.

| method | epoch | termination | FID↓ | sFID↓ | IS↑ |
|---|---|---|---|---|---|
| SiT-XL/2 | 600 | - | 2.62 | 4.18 | 252.2 |
| +REPA | 80 | - | 2.55 | **4.16** | 241.2 |
| +REPA | 100 | - | 2.36 | **4.16** | **254.2** |
| **+HASTE** | 80 | - | 2.78 | 4.34 | 209.5 |
| **+HASTE** | 80 | 60 | 2.49 | 4.20 | 231.4 |
| **+HASTE** | 100 | 60 | **2.34** | 4.23 | 253.4 |

Table 13: Evaluation results on ImageNet $512 \times 512$. ↑ and ↓ denote higher and lower values are better, respectively. **Bold font** denotes the best performance.

We report the quantitative results in Table 13, which reflects that HASTE can effectively accelerates the training process with various image resolutions, also validating our alignment and termination strategy. We evaluate the performance of SiT-XL/2+HASTE with SDE sampler (NFEs = 250, cfg scale $w = 1.25$) and do not apply guidance interval [28].

## A.5 Detailed Quantitative Results

We provide detailed evaluation results of HASTE on different SiT models in Table 14. All results are reported with the SDE Euler-Maruyama sampler (NFEs = 250) and without classifier-free guidance.

| model | #params | iteration | FID↓ [16] | sFID↓ [35] | IS↑ [42] | Prec.↑ [27] | Rec.↑ [27] |
|---|---|---|---|---|---|---|---|
| SiT-B/2 [34] | 130M | 400K | 33.0 | 6.46 | 43.7 | 0.53 | 0.63 |
| **+HASTE** | 130M | 100K | 39.9 | 7.16 | 35.8 | 0.52 | 0.61 |
| **+HASTE** | 130M | 200K | 25.7 | 6.66 | 57.0 | 0.59 | 0.62 |
| **+HASTE** | 130M | 400K | **19.6** | **6.38** | **73.0** | **0.62** | **0.64** |
| SiT-L/2 [34] | 458M | 400K | 18.8 | 5.29 | 72.0 | 0.64 | 0.64 |
| **+HASTE** | 458M | 100K | 19.6 | 5.70 | 67.9 | 0.64 | 0.63 |
| **+HASTE** | 458M | 200K | 12.1 | 5.28 | 96.1 | 0.68 | 0.64 |
| **+HASTE** | 458M | 400K | **8.9** | **5.18** | **118.9** | **0.69** | **0.66** |
| SiT-XL/2 [34] | 675M | 7M | 8.6 | 6.32 | 131.7 | 0.68 | 0.67 |
| **+HASTE** | 675M | 100K | 15.9 | 5.64 | 78.1 | 0.67 | 0.62 |
| **+HASTE** | 675M | 200K | 9.9 | 5.04 | 108.8 | 0.69 | 0.64 |
| **+HASTE** | 675M | 250K | 8.4 | 4.90 | 119.6 | 0.70 | **0.65** |
| **+HASTE** | 675M | 400K | 7.3 | 5.05 | 128.7 | 0.72 | 0.64 |
| **+HASTE** | 675M | 500K | **5.3** | **4.72** | **148.5** | **0.73** | **0.65** |

Table 14: Additional evaluation results on ImageNet 256 × 256. ↑ and ↓ denote higher and lower values are better, respectively. **Bold font** denotes the best performance.

Additionally, we provide the results of SiT-XL/2+HASTE with different classifier-free guidance [17] scales and intervals [28].

| model | #params | iteration | interval | CFG scale | FID↓ | sFID↓ | IS↑ | Prec.↑ | Rec.↑ |
|---|---|---|---|---|---|---|---|---|---|
| SiT-XL/2 | 675M | 7M | [0, 1] | 1.50 | 2.06 | 4.50 | 270.3 | **0.82** | 0.59 |
| **+HASTE** | 675M | 500K | [0, 1] | 1.25 | 2.18 | 4.67 | 240.4 | 0.81 | 0.60 |
| **+HASTE** | 675M | 500K | [0, 0.7] | 1.50 | 1.80 | 4.58 | 252.1 | 0.80 | 0.61 |
| **+HASTE** | 675M | 500K | [0, 0.6] | 1.825 | 1.74 | 4.74 | 268.7 | 0.80 | 0.62 |
| **+HASTE** | 675M | 2M | [0, 0.7] | 1.7 | 1.45 | 4.55 | 297.3 | 0.80 | 0.64 |
| **+HASTE** | 675M | 2M | [0, 0.7] | 1.65 | 1.44 | 4.56 | 289.4 | 0.79 | 0.64 |
| **+HASTE** | 675M | 2M | [0, 0.7] | 1.675 | 1.44 | 4.55 | 293.7 | 0.80 | 0.64 |
| **+HASTE** | 675M | 2.5M | [0, 0.7] | 1.7 | 1.43 | 4.56 | 298.8 | 0.80 | 0.64 |
| **+HASTE** | 675M | 2.5M | [0, 0.7] | 1.65 | 1.43 | 4.57 | 290.7 | 0.80 | 0.64 |
| **+HASTE** | 675M | 2.5M | [0, 0.72] | 1.65 | 1.42 | **4.49** | **299.5** | 0.80 | **0.65** |
| **+HASTE** | 675M | 3M | [0, 0.7] | 1.67 | **1.41** | 4.51 | 296.9 | 0.80 | **0.65** |

Table 15: Evaluation results on ImageNet 256 × 256 with different classifier-free guidance settings.

# B  Additional Implementation Details.

| | SiT-B | SiT-L | SiT-XL | DiT-XL |
|---|---|---|---|---|
| **Architecture** | | | | |
| input dim. | 32×32×4 | 32×32×4 | 32×32×4 | 32×32×4 |
| num. layers | 12 | 24 | 28 | 28 |
| hidden dim. | 768 | 1024 | 1152 | 1152 |
| num. heads | 12 | 16 | 16 | 16 |
| **HASTE** | | | | |
| $\lambda_R$ | 0.5 | 0.5 | 0.5 | 0.5 |
| $\lambda_A$ | 0.5 | 0.5 | 0.5 | 0.5 |
| alignment depth | 5 | 8 | 8 | 8 |
| student layers | [2, 3, 4] | [4, 5, 6, 7] | [4, 5, 6, 7] | [4, 5, 6, 7] |
| teacher model | DINOv2-B [36] | DINOv2-B [36] | DINOv2-B [36] | DINOv2-B [36] |
| teacher layers | [7, 9, 11] | [8, 9, 10, 11] | [8, 9, 10, 11] | [8, 9, 10, 11] |
| termination iter. | 100 K | 250 K | 250 K | 250 K |
| alignment heads | 0-11 | 0-11 | 0-11 | 0-11 |
| **Optimization** | | | | |
| batch size | 256 | 256 | 256 | 256 |
| optimizer | AdamW [25, 33] | AdamW [25, 33] | AdamW [25, 33] | AdamW [25, 33] |
| lr | 0.0001 | 0.0001 | 0.0001 | 0.0001 |
| $(\beta_1, \beta_2)$ | (0.9, 0.999) | (0.9, 0.999) | (0.9, 0.999) | (0.9, 0.999) |
| weight decay | 0 | 0 | 0 | 0 |
| **Diffusion** | | | | |
| objective | linear interpolants | linear interpolants | linear interpolants | improved DDPM |
| prediction | velocity | velocity | velocity | noise and variance |
| sampler | Euler-Maruyama | Euler-Maruyama | Euler-Maruyama | Euler-Maruyama |
| sampling steps | 250 | 250 | 250 | 250 |

Table 16: Detailed training settings.

**Further implementation details.** For XL and L-sized models, we set the feature alignment depth to 8 following REPA, and extract the attention maps from layer [4, 5, 6, 7] (counting from 0) of diffusion transformers, to align with those from layer [8, 9, 10, 11] of DINOv2-B. According to [31], the performance almost saturates when transferring 12 out of 16 heads, and the student can also develop its own attention patterns for unused heads. Specifically, since the number of heads for DINOv2-B layer is only 12, we conduct attention alignment partially over the first 12 heads of diffusion transformer layer. For B-sized models, the feature alignment depth is adjusted to 5, and we extract the attention maps from layer [2, 3, 4] to align with those from layer [7, 9, 11] of DINOv2-B.

We enable mixed-precision (fp16) for efficient training. For data pre-processing, we leverage the protocols provided in EDM2 [21] to pre-compute latent vectors from images with stable diffusion VAE [40]. Specifically, we use `stabilityai/sd-vae-ft-ema` decoder to translate generated latent vectors into images. Following REPA [55], we also use three-layer MLP with SiLU activations [7] as the projector of hidden states. For MM-DiT, we use CLIP [38] text model to encode captions.

## C HASTE Training Loop

For clarity, we provide pseudocode in Algorithm 1 for the HASTE training loop with a fixed iteration as the trigger to further illustrate our two-phase schedule in addition to Figure 2.

---

**Algorithm 1** HASTE Training Loop

---

**Require:** Diffusion model $G_\theta$, Pre-trained encoder $E$
**Require:** Training dataset $\mathcal{D}$, Termination point $\tau$
**Require:** Loss weights $\lambda_R, \lambda_A$, Maximum iterations $N$
**Ensure:** Trained diffusion model $G_\theta$

1: **for** $i = 1$ to $\tau$ **do**        ▷ **Phase I: Holistic Alignment**
2:     Sample $x, t, \epsilon$ from $\mathcal{D}, \mathcal{U}(0,1), \mathcal{N}(0,I)$      ▷ Sample training batch
3:     $x_t \leftarrow \alpha_t x + \beta_t \epsilon$
4:     $h_t \leftarrow \text{Hidden}(G_\theta, x_t, t)$       ▷ DiT hidden states
5:     $A_G \leftarrow \text{Attn}(G_\theta, x_t, t)$       ▷ DiT attention maps
6:     $y \leftarrow E(x)$       ▷ Encoder features
7:     $A_E \leftarrow \text{Attn}(E, x)$       ▷ Encoder attention maps
8:     $\mathcal{L}_{\text{diff}} \leftarrow \text{MSE}(G_\theta(x_t, t), target)$
9:     $\mathcal{L}_{\text{REPA}} \leftarrow -\text{CosSim}(y, \text{Proj}(h_t))$
10:    $\mathcal{L}_{\text{ATTA}} \leftarrow \text{CrossEntropy}(A_G, A_E)$
11:    $\mathcal{L} \leftarrow \mathcal{L}_{\text{diff}} + \lambda_R \mathcal{L}_{\text{REPA}} + \lambda_A \mathcal{L}_{\text{ATTA}}$       ▷ Alignment and denoising
12:    $\theta \leftarrow \theta - \eta \nabla_\theta \mathcal{L}$       ▷ Update parameters
13: **end for**

14: **for** $i = \tau + 1$ to $N$ **do**        ▷ **Phase II: Pure Denoising**
15:    Sample $x, t, \epsilon$ from $\mathcal{D}, \mathcal{U}(0,1), \mathcal{N}(0,I)$      ▷ Sample training batch
16:    $x_t \leftarrow \alpha_t x + \beta_t \epsilon$
17:    $\mathcal{L} \leftarrow \text{MSE}(G_\theta(x_t, t), target)$       ▷ Denoising only
18:    $\theta \leftarrow \theta - \eta \nabla_\theta \mathcal{L}$       ▷ Update parameters
19: **end for**

---

## D Additional Visualizations

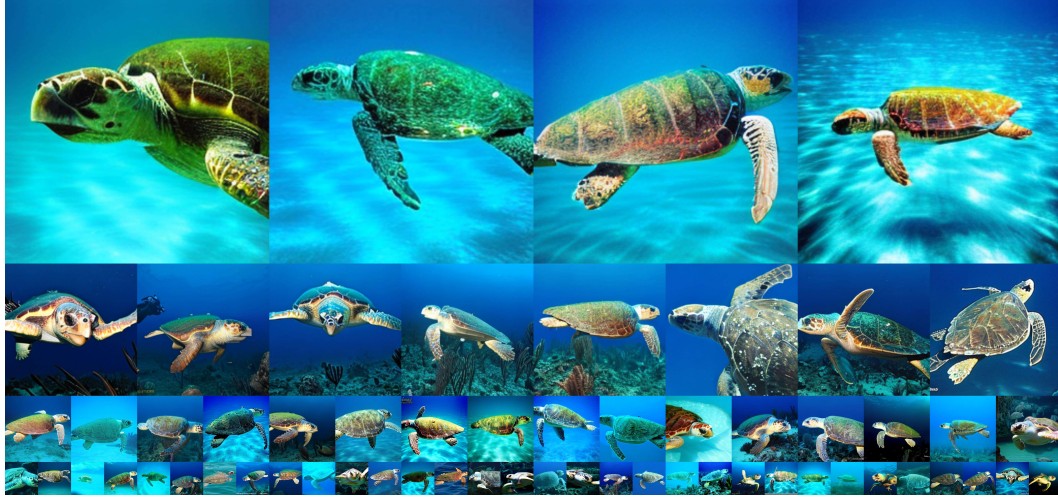

Figure 10: Uncurated generation results of SiT-XL/2+HASTE. We use classifier-free guidance with $w = 4.0$. Class label = "loggerhead sea turtle" (33).

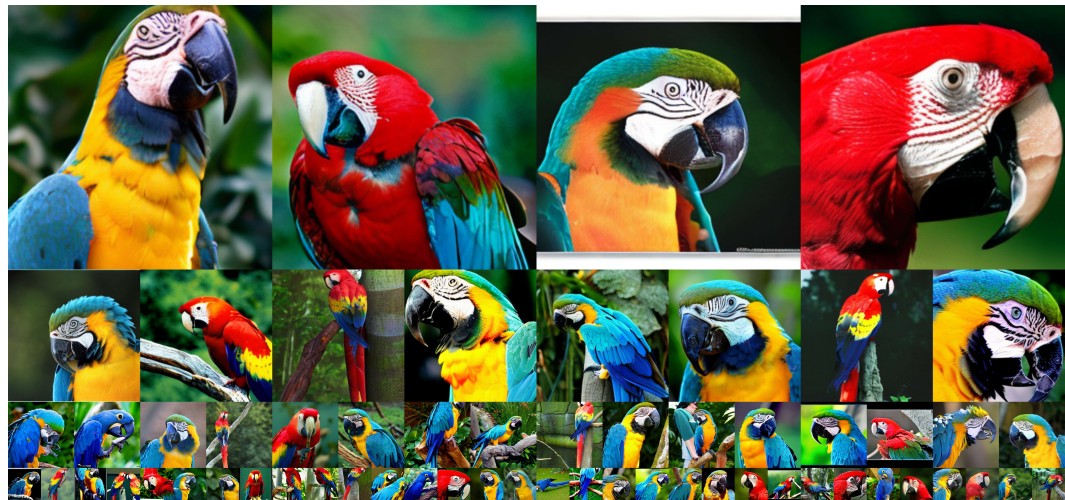

Figure 11: Uncurated generation results of SiT-XL/2+HASTE. We use classifier-free guidance with $w = 4.0$. Class label = "macaw" (88).

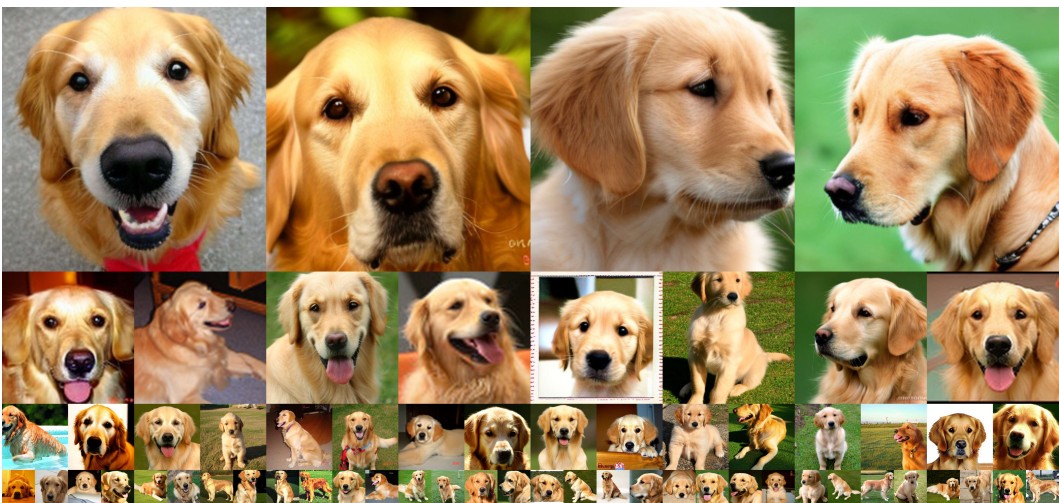

Figure 12: Uncurated generation results of SiT-XL/2+HASTE. We use classifier-free guidance with $w = 4.0$. Class label = "golden retriever" (207).

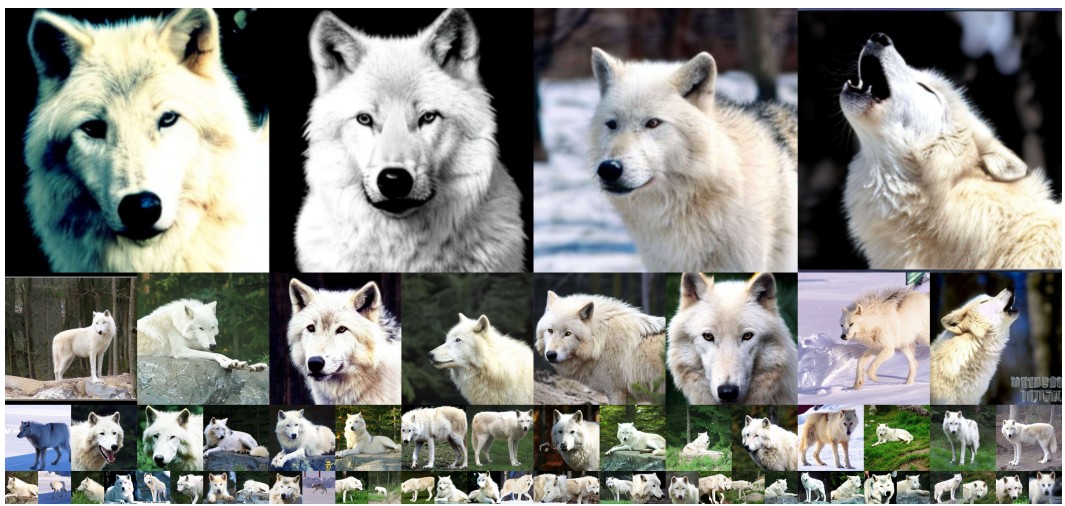

Figure 13: Uncurated generation results of SiT-XL/2+HASTE. We use classifier-free guidance with $w = 4.0$. Class label = "arctic wolf" (270).

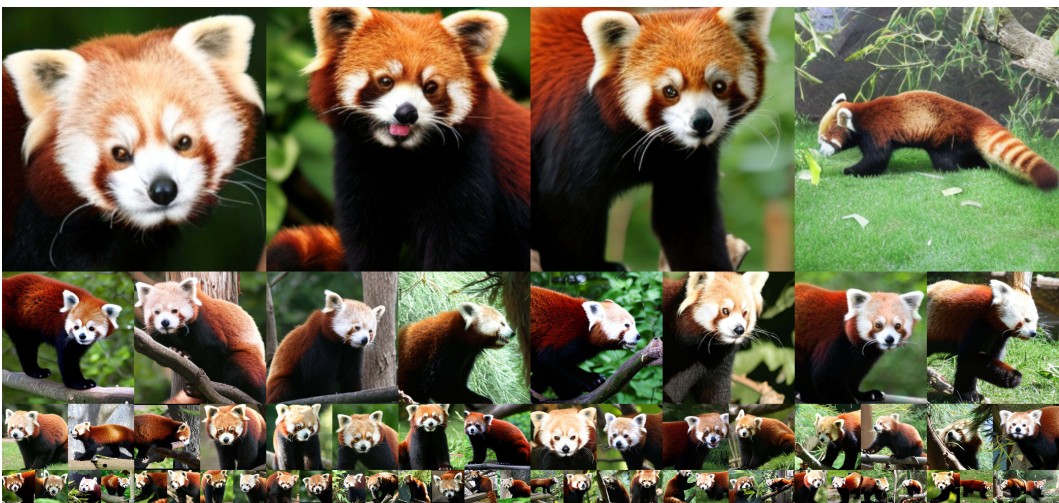

Figure 14: Uncurated generation results of SiT-XL/2+HASTE. We use classifier-free guidance with $w = 4.0$. Class label = "red panda" (387).

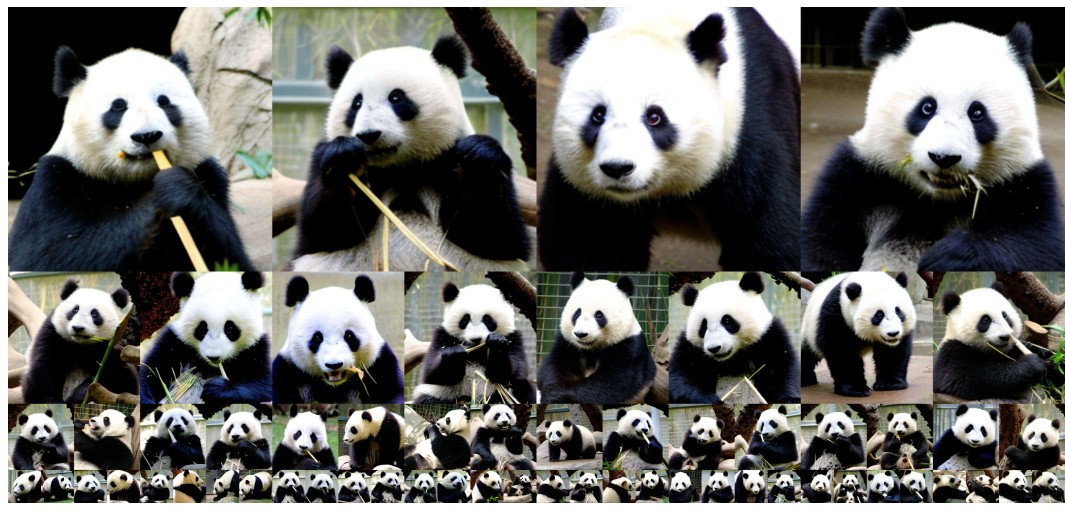

Figure 15: Uncurated generation results of SiT-XL/2+HASTE. We use classifier-free guidance with $w = 4.0$. Class label = "panda" (388).

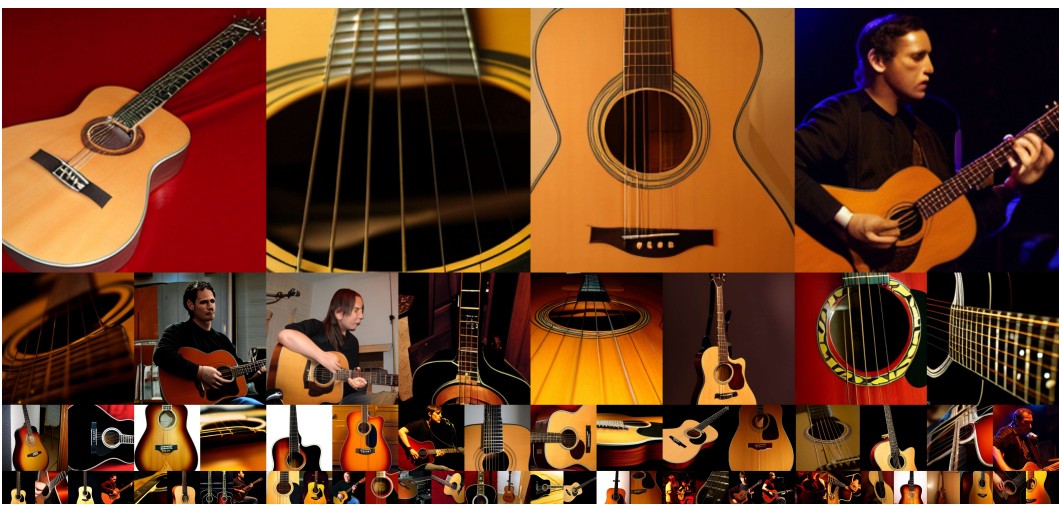

Figure 16: Uncurated generation results of SiT-XL/2+HASTE. We use classifier-free guidance with $w = 4.0$. Class label = "acoustic guitar" (402).

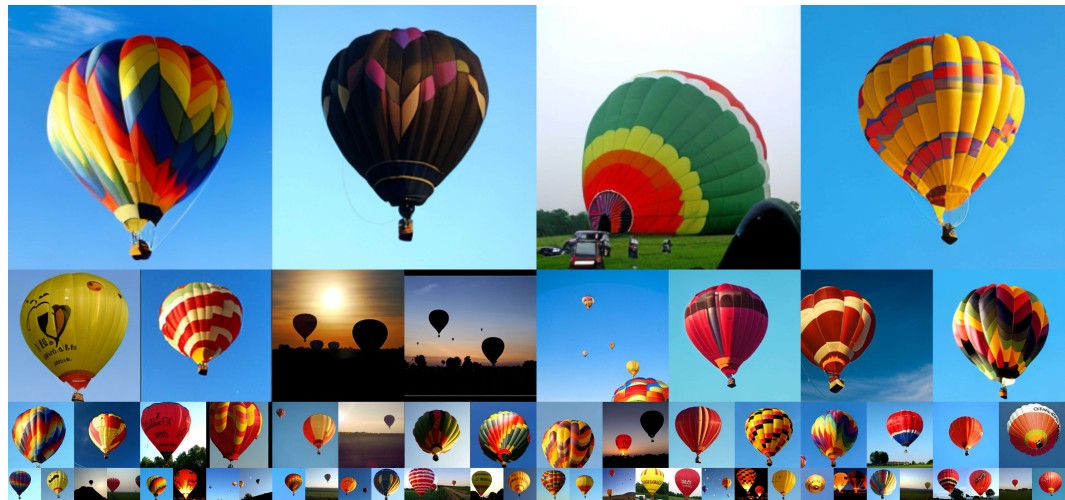

Figure 17: Uncurated generation results of SiT-XL/2+HASTE. We use classifier-free guidance with $w = 4.0$. Class label = "balloon" (417).

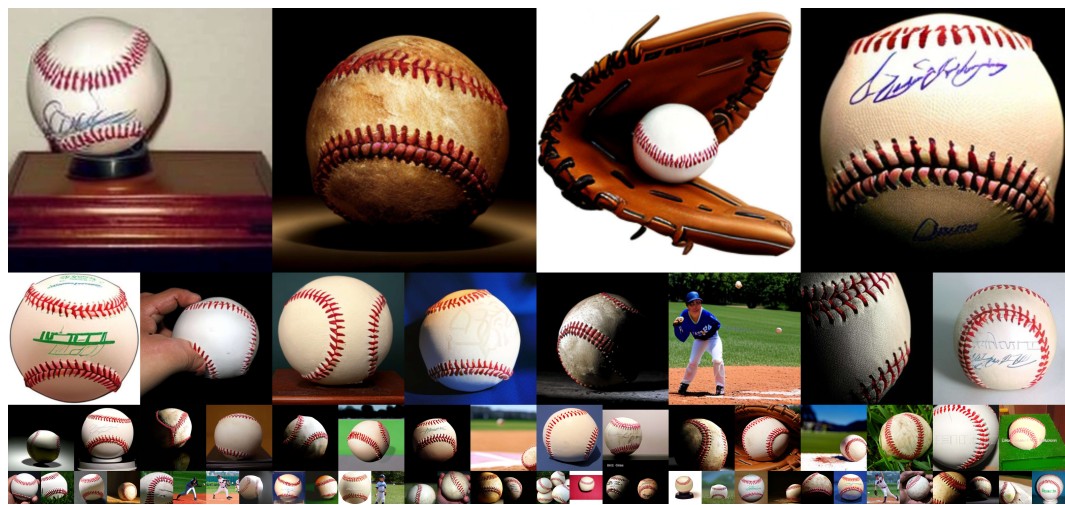

Figure 18: Uncurated generation results of SiT-XL/2+HASTE. We use classifier-free guidance with $w = 4.0$. Class label = "baseball" (429).

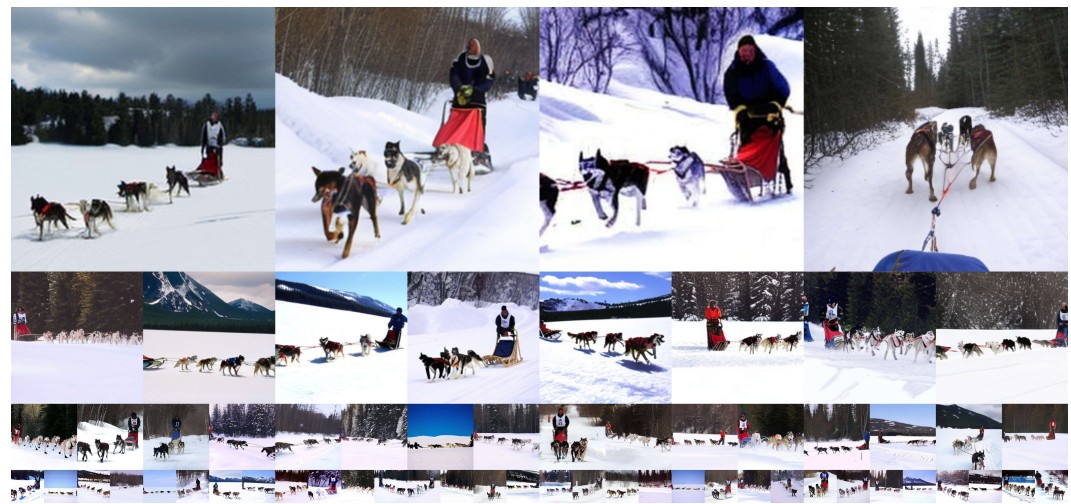

Figure 19: Uncurated generation results of SiT-XL/2+HASTE. We use classifier-free guidance with $w = 4.0$. Class label = "dog sled" (537).

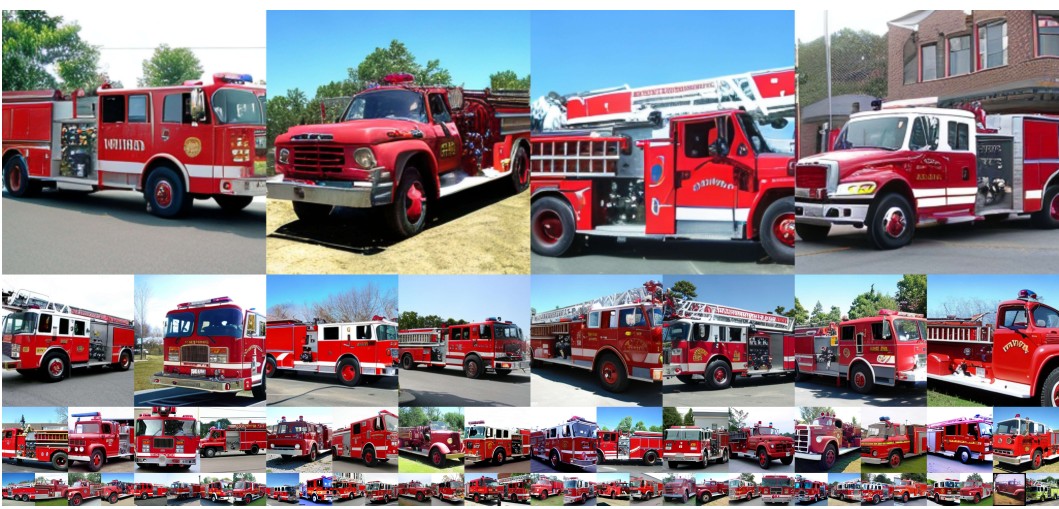

Figure 20: Uncurated generation results of SiT-XL/2+HASTE. We use classifier-free guidance with $w = 4.0$. Class label = "fire truck" (555).

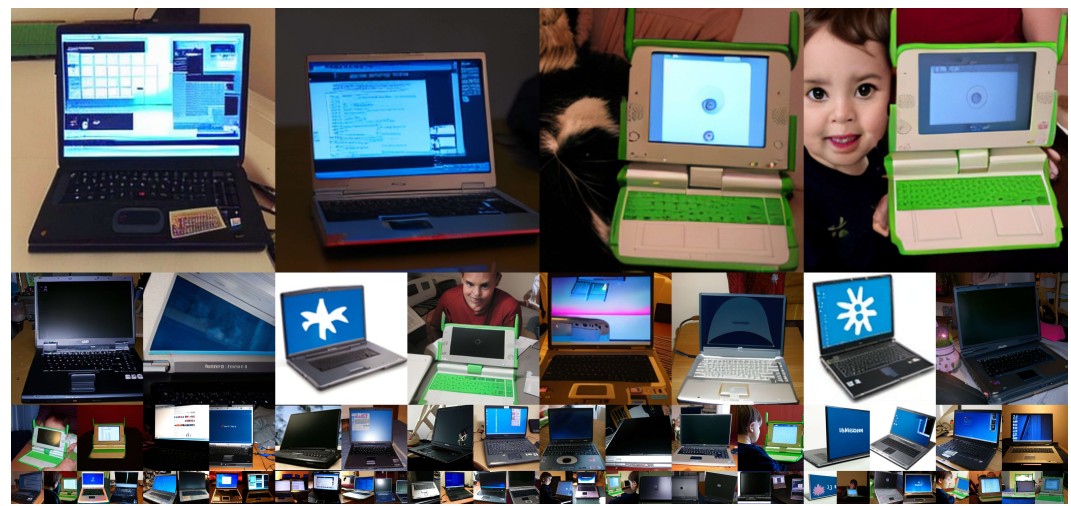

Figure 21: Uncurated generation results of SiT-XL/2+HASTE. We use classifier-free guidance with $w = 4.0$. Class label = "laptop" (620).

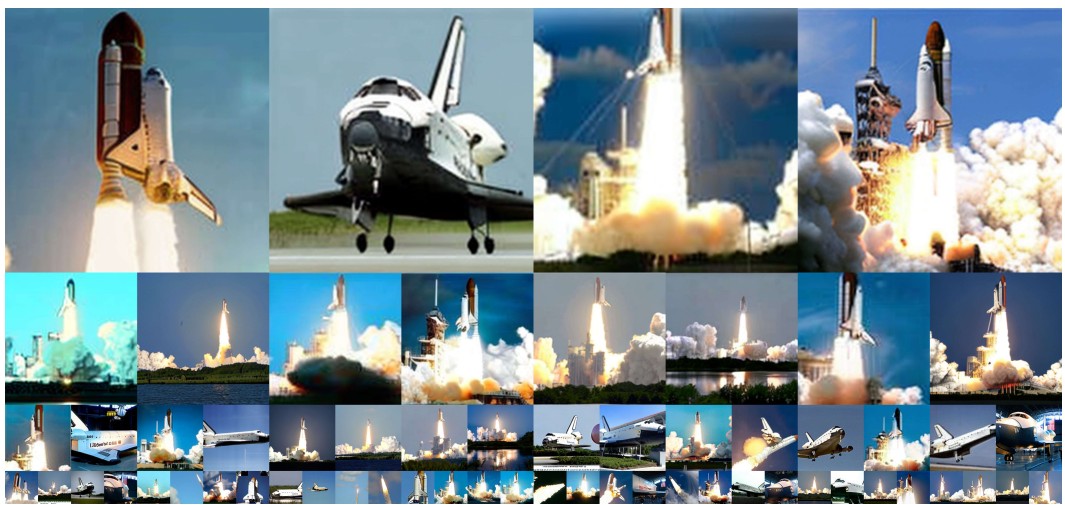

Figure 22: Uncurated generation results of SiT-XL/2+HASTE. We use classifier-free guidance with $w = 4.0$. Class label = "space shuttle" (812).

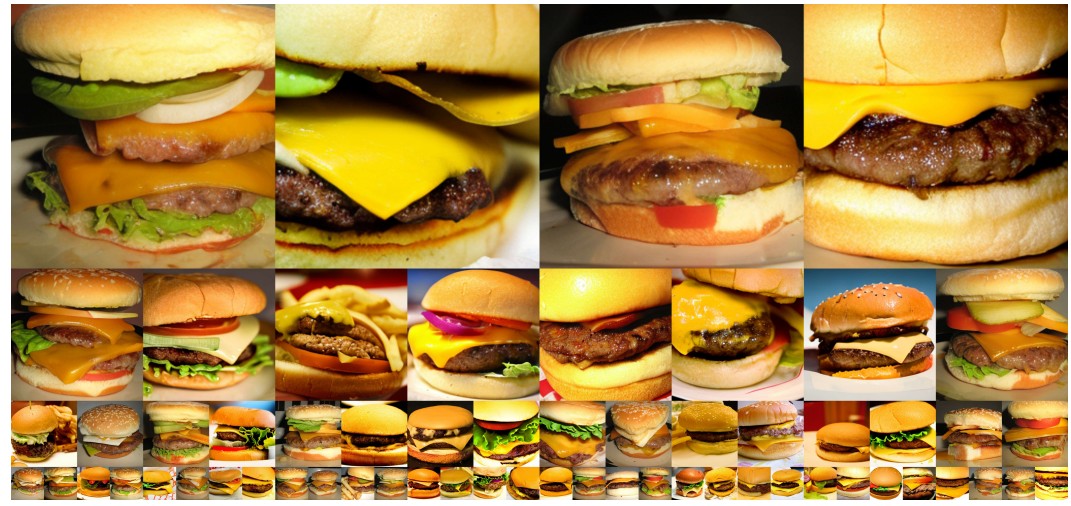

Figure 23: Uncurated generation results of SiT-XL/2+HASTE. We use classifier-free guidance with $w = 4.0$. Class label = "cheeseburger" (933).

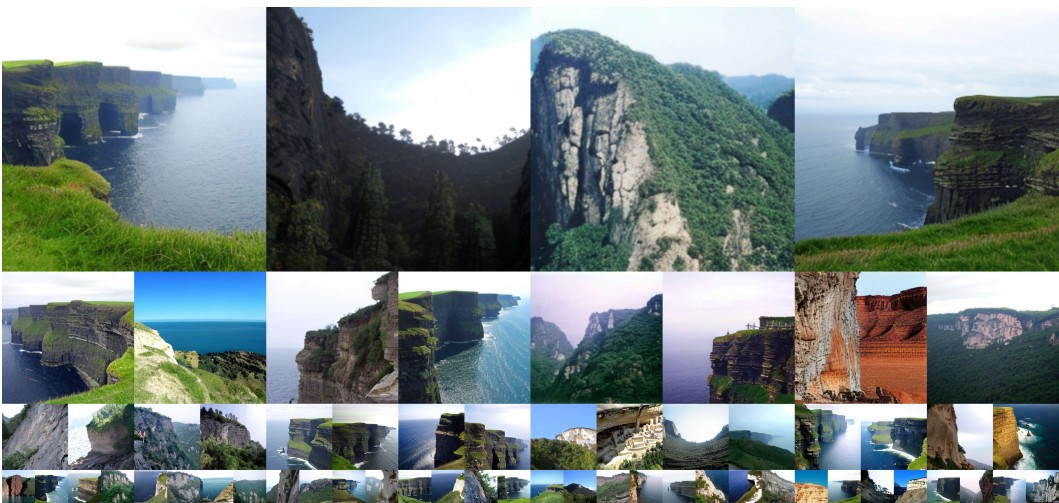

Figure 24: Uncurated generation results of SiT-XL/2+HASTE. We use classifier-free guidance with $w = 4.0$. Class label = "cliff drop-off" (972).

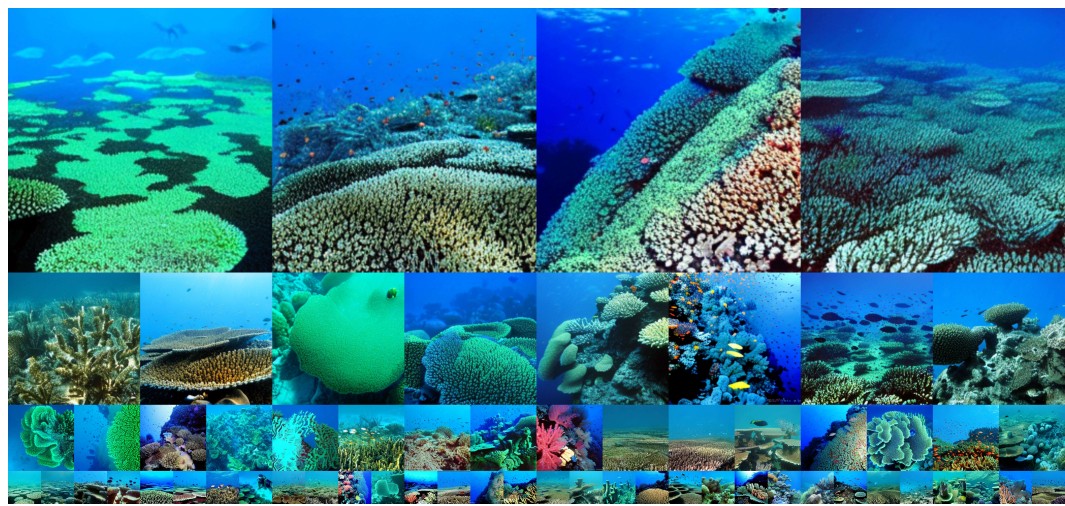

Figure 25: Uncurated generation results of SiT-XL/2+HASTE. We use classifier-free guidance with $w = 4.0$. Class label = "coral reef" (973).

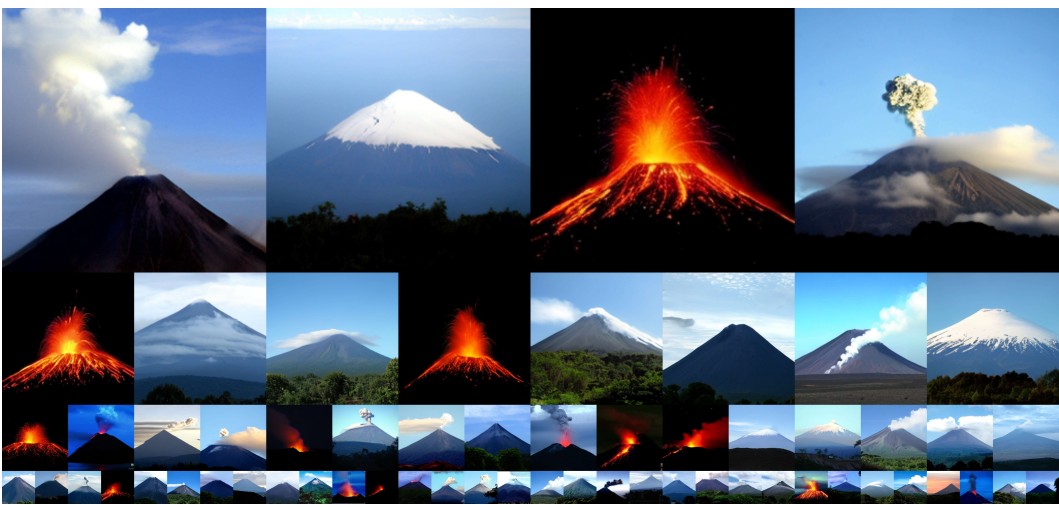

Figure 26: Uncurated generation results of SiT-XL/2+HASTE. We use classifier-free guidance with $w = 4.0$. Class label = "volcano" (980).

