# OpenReview forum: "REPA Works Until It Doesn’t: Early-Stopped, Holistic Alignment Supercharges Diffusion Training"
_NeurIPS.cc/2025/Conference — NeurIPS 2025 poster_

### Official Review · Reviewer_FRQ1 · 2025-06-24

**Clarity:** 2
**Significance:** 3
**Originality:** 3
**Rating:** 3
**Confidence:** 4

**Summary:**

This paper identifies limitations in REPA, a regularization method aimed at accelerating diffusion transformer training, and proposes simple improvements. The authors show that REPA is only helpful in the early stages of training and may become detrimental later, advocating for an early stopping mechanism that disables REPA in the middle of the training. Furthermore, they introduce a holistic alignment, which aligns not only the features of DINO and the diffusion transformer (as in REPA) but also their attention maps. The paper empirically demonstrates the benefits of both REPA early stopping and attention alignment through ImageNet generation experiments.

**Questions:**

In Table 4, it's unclear why the performance of SiT-XL/2 at 500K iterations is worse than at 400K. This inconsistency is not addressed.

**Ethical Concerns:**

["NO or VERY MINOR ethics concerns only"]

**Final Justification:**

This paper shows that the REPA regularization, which is effective for training diffusion models, is only beneficial in the early iterations of training, and proposes a simple method to address this limitation.

However, the “gradient trigger” component of the method is not actually used in practice, yet it occupies a substantial portion of the method description, which may confuse readers. The authors explain that it can be used for analysis purposes, but in the current manuscript, it is unclear whether the gradient trigger is merely for analysis or is an effective part of the method. This gives the paper an immature impression in this regard.

Therefore, I lower my score to 3. Nonetheless, the paper’s core observation is meaningful, and I will not oppose the accept recommendation.

**Limitations:**

In the limitations section, the paper should explicitly state that the use of external pre-trained models like DINO requires additional pre-training computation and introduces extra computational overhead during diffusion model training.

**Paper Formatting Concerns:**

The paper would benefit from a figure that illustrates the REPA and ATTA regularization mechanisms.

**Quality:**

2

**Strengths And Weaknesses:**

## Strengths

- Clearly identifies and analyzes the limitations of REPA, offering valuable insights for the generative AI community.

- Proposes attention alignment (ATTA), a novel idea not yet explored in diffusion model literature.

- Demonstrates faster convergence compared to DiT, SiT, and REPA baselines.

## Weaknesses

- The paper is poorly written, with missing explanations in several figures. For instance, Figure 5 does not clarify what [0, 1], [0.25, 1], etc., mean, nor how low-frequency information is filtered. In Figure 6 (b and c), the lack of baselines makes it difficult to interpret the quality of the results.

- While the paper shows that REPA mainly helps with learning low-frequency semantic information early in training, it overlooks a simpler baseline: timestep sampling biased toward high noise levels, as in EDM [A, B] and Stable Diffusion 3 [C]. If learning low-frequency semantics early is beneficial, allocating more training computation to larger timesteps (which carry low-frequency information) may serve as a stronger and more straightforward baseline than using external vision encoders.

- While Line 190 mentions the use of a gradient-angle trigger as an alternative to fixed-iteration early stopping, it appears that this trigger is not actually used in the experiments.

[A] Karras et al., Elucidating the Design Space of Diffusion-Based Generative Models, NeurIPS 2022.

[B] Karras et al., Analyzing and Improving the Training Dynamics of Diffusion Models, CVPR 2024.

[C] Esser et al., Scaling Rectified Flow Transformers for High-Resolution Image Synthesis, ICML 2024.

---

> ### Author Rebuttal · Authors · 2025-07-31
>
> Thanks for your comments. We discuss the questions as follows.
>
> > **Q1:** Missing explanations in several figures.
>
> Thank you for your comment. We acknowledge that the illustrations of Figures 5,6 can be clearer. Therefore, we provide detailed explanations of them.
>
> - **Figure 5:** The time interval [0, 0.25] stands for the diffusion timesteps where the input to pre-trained encoders is replaced with low-frequecy part of the image. For image filtering, we convert the image to frequency domain using the Fourier transform, and filter out the high-frequency components, then invertly convert it back to an image.
> - **Figure 6:** We compute the feature and attention alignment progress mainly to explain the complementary effect of both methods. We will add baselines to (b) and (c) in the latest version.
>
> > **Q2:** Will allocating more training computation to larger timesteps with higher noise levels serve as a better baseline?
>
> Thanks for your insightful question. We provide comparision of REPA, HASTE, and log-normal timestep sampling on SiT-B/2.
>
> | method         | iteration | FID$\downarrow$ | sFID$\downarrow$ | IS$\uparrow$ |
> | - | - | - | - | - |
> | SiT+log-normal | 50K       | 92.27           | **11.11**        | 14.76        |
> | SiT+REPA       | 50K       | 78.20           | 11.71            | 17.1         |
> | SiT+HASTE      | 50K       | **71.32**       | 13.85            | **19.26**    |
>
> While log-normal has been proved effective on models like EDM, we find acceleration methods based on external representations still achieve better performance.
>
> > **Q3:** The use of gradient-angle trigger.
>
> Thanks for your important question. In the following, we first explain the role of gradient signals in our method, and then provide a plain example of automatic gradient-angle trigger.
> - **Dual termination strategy clarification:** Our method employs a hybrid approach where fixed iterations serve as the primary and practical termination strategy, while the gradient-angle rule provides theoretical insight and robustness across different datasets. Specifically:
>     - In practice, we find setting a fixed termination iteration $\tau$ for HASTE actually performs reliably well in ImageNet and MS-COCO experiments. Gradient-angle threshold serves as a supplemental alternative, adding adaptability to different scenerios.
>     - The gradient analysis helps us to locate the termination point. As we state in Lines 283-287, we choose to stop around 250K iteration when we observe limited alignment benefits and relatively low gradient cosine similarities.
> - **Plain implementation of automatic gradient-angle trigger:** We train SiT-B/2 on ImageNet 256 $\times$ 256 following original settings of HASTE using a gradient-angle trigger.
>     - **Implementation:**
>         - We start evaluating the gradient cosine similarity between holistic alignment and denoising objectives every 10K iterations from 50K iteration, when the training process has been relatively stable.
>         - In each round of evaluation, we compute the similarity separately on selected diffusion timesteps $t$ at different noise levels with 2048 images randomly sampled from ImageNet, and take the average value. When the average similarity falls below a threshold $\delta$, we terminate the holistic alignment.
>         - In the plain example, we perform gradient evaluation on the 5th SiT block (counting from 1). We select $t=0.02$, $t=0.1$, $t=0.5$ and $t=0.8$, and set $\delta=-0.05$.
>     - **Results:**
>         - **Gradient cosine similarity:**
>
>         |timestep\iteration|50K|60K|70K|80K|
>         |-|-|-|-|-
>         |0.02|-0.0174|0.0418|0.0872|-0.0032|
>         |0.1|0.0126|-0.0767|-0.0148|-0.1155|
>         |0.5|0.0134|0.0106|-0.0848|-0.1272|
>         |0.8|0.0034|0.0406|0.1025|-0.0212|
>         |average|0.0030|0.0041|0.0225|**-0.0668(terminate)**|
>         - **Performance:**
>
>         |method|iteration|termination|FID$\downarrow$|sFID$\downarrow$|IS$\uparrow$|
>         |-|-|-|-|-|-|
>         |SiT|100K|-|63.46|7.31|20.6|
>         |SiT+HASTE|100K|-|39.86|7.16|35.8|
>         |SiT+HASTE|100K|80K|**38.69**|**6.88**|**36.4**|
>
>     - **Findings:** Our automatic trigger terminates the holistic alignment at 80K iteration, and achieves better performance at 100K iteration.
>     - **Limitations:** Though we validate the automatic gradient-angle trigger on SiT-B/2, we find it challenging to design such trigger for precise termination, because of:
>         - **Additional overhead:** Gradient evaluation introduces extra computational cost, limiting the practical evaluation frequency.
>         - **Hyperparameter setting:** The automatic trigger is sensitive to hyperparameters, such as block choice, timestep selection, and the threshold.
>     - **Conclusion:** Given that monitoring model performance during training is a standard practice in modern deep learning workflows, we think **it's more practical to integrate the periodic evaluation of both metrics and gradient cosine similarity to determine the optimal termination point**.
> > **Q4:** Inconsistent performance of SiT-XL/2 at 400K and 500K iteration.
>
> Thanks for your question. As we mentioned in Line 277-281, we attribute the inconsistency to over-regularization introduced by continuously using holistic alignment. Actually, our termination strategy at 250K iteration helps to alleviate such a trend and demonstrates better performance at 500K iteration.
>
> > **Q5:** The use of external pre-trained models introduces extra training cost.
>
> Thank you for the question. We evaluate the extra training overhead of HASTE on SiT-XL/2 introduced by external vision model DINOv2-B:
>
> - **Extra memory overhead:** The peak CUDA memory with local batchsize 32:
>
> 	| HASTE       | REPA        |
> 	| -| - |
> 	| 30666.84 MB | 27818.01 MB |
>
> - **DINOv2-B forward overhead:** During training, we compute DINOv2‑B representations online following REPA. The forward pass of DINOv2‑B incurs approximately 23.19 GFLOPs for the base‑size model at 224×224 input resolution without XFormers.
>
> We will add these evaluations in our latest version.
>
> > **Formatting Concern:** "The paper would benefit from a figure that illustrates the REPA and ATTA regularization mechanisms."
>
> Thanks for your advice. We will add the illustrations in the latest version.

---

> > ### Comment · Reviewer_FRQ1 · 2025-08-04
> >
> > After reading the rebuttal, I still have a few questions.
> >
> > Regarding Q2, I would like to clarify what mean and variance were used in the log-normal distribution. The reason I originally asked about the log-normal case was that it corresponds to training with a log-normal distribution biased toward larger timesteps with higher noise levels. However, based on the current table alone, it is not clear whether the log-normal baseline indeed concentrates on larger timesteps.
> >
> > Regarding Q3, I am wondering whether the initial manuscript actually employed the gradient-angle trigger, or if it was only mentioned as a possible approach without being used in practice. In addition, I would like to know the criteria for selecting the specific values of t=0.02, 0.1, 0.5, 0.8.

---

> > > ### Author Response · Authors · 2025-08-04
> > >
> > > > **Regarding Q2: Clarification of Log-Normal Distribution Parameters**
> > >
> > > Thank you for your follow-up regarding the log-normal baseline and timestep sampling.
> > >
> > > **Clarification:**
> > > - In our implementation, we sample $\sigma$ using the formula $\sigma = e^z$, where $z \sim \mathcal{N}(0, 1)$.
> > > - This sampled $\sigma$ is then mapped to the timestep domain via $t = \frac{\sigma}{1+\sigma}$.
> > >
> > > **Interpretation:**
> > > - This parameterization **places greater sampling weight on intermediate timesteps** compared to earlier ones.
> > >
> > > **Context:**
> > > - Non-uniform timestep sampling schemes—such as this log-normal variant—have been shown to be effective in previous works (e.g., EDM [1,2], Stable Diffusion 3 [3]).
> > > - However, **we find that leveraging external pre-trained vision encoders yields greater acceleration and improved performance** over timestep reweighting strategies alone.
> > >
> > > Given our main research goal is to investigate the acceleration benefits brought by pre-trained encoders, we have **focused our experiments on methods that integrate external models**, with log-normal sampling serving as a baseline rather than the main emphasis.
> > >
> > > [1]. Karras et al., *Elucidating the Design Space of Diffusion-Based Generative Models*, NeurIPS 2022.
> > >
> > > [2]. Karras et al., *Analyzing and Improving the Training Dynamics of Diffusion Models*, CVPR 2024.
> > >
> > > [3]. Esser et al., *Scaling Rectified Flow Transformers for High-Resolution Image Synthesis*, ICML 2024.
> > >
> > >
> > > > **Regarding Q3: Actual Usage and Design of the Gradient-Angle Trigger**
> > >
> > > Thank you for highlighting these important points.
> > >
> > > **Practical Use in Experiments:**
> > > - In our current experiments, **we primarily employ a fixed iteration termination for HASTE**, and use gradient analysis as a *diagnostic* tool rather than as an active stopping criterion during training.
> > > - The **gradient-angle trigger** was suggested as a promising direction, but **was not actively deployed as the primary termination method** due to practical considerations.
> > >
> > > **Limitations of the Gradient Trigger:**
> > > - We have observed several challenges that limit the practicality of the automatic gradient trigger:
> > >     - **Computational overhead:** Evaluating gradients at regular intervals requires additional computation, potentially slowing overall training.
> > >     - **Sensitivity to hyperparameters:** The efficacy of the trigger depends on carefully choosing evaluation block, timesteps, and the threshold $\delta$.
> > > - For these reasons, **an iteration-based stopping rule currently offers greater reliability and simplicity** in our training regime.
> > >
> > > **Criteria for Timestep Selection in Gradient Evaluation:**
> > > - To provide a balanced measure of alignment progress, we select **representative timesteps from a range of noise levels**:
> > >     - **$t=0.02$ and $t=0.1$**: Represent lower noise levels, reflecting *fine-grained* generation stages.
> > >     - **$t=0.5$ and $t=0.8$**: Represent higher noise levels, probing *coarser* generation steps.
> > > - This approach aims to capture **alignment across both the early (fine) and later (coarse) stages** of the diffusion process.
> > >
> > > **Outlook:**
> > > - We agree that the development of more robust and efficient automatic termination triggers—possibly guided by gradient-based signals—**remains an open and promising direction** for future work, and we appreciate the opportunity to clarify these points.
> > >
> > > **Summary of Actions:**
> > > - We will add explicit clarifications regarding both the role and limitations of gradient-angle triggers, and detail our timestep selection criteria in the revised manuscript for improved transparency.

---

> > > > ### Comment · Reviewer_FRQ1 · 2025-08-05
> > > >
> > > > Regarding Q2, it seems that a baseline focusing the weight on the “larger timesteps” — which the authors identified as important — rather than on intermediate timesteps would have been necessary. In other words, the mean of z hould have been a positive value such as 1 or 3, instead of 0.
> > > >
> > > > Regarding Q3, it appears that the gradient-angle trigger was rarely used in the paper and does not seem to be an effective solution. While it may help in understanding the behavior of REPA, it also requires a hyperparameter search. Therefore, I believe the truly meaningful component of this work is the early stopping of REPA regularization, which somewhat weakens the contribution of the paper.
> > > >
> > > > Since the concerns about the gradient-angle trigger were raised also by two other reviewers, I consider this to be a rather significant issue.

---

> > > > > ### Author Response · Authors · 2025-08-06
> > > > >
> > > > > Thanks for your follow-up. We discuss the questions as follows.
> > > > >
> > > > > > **Regarding Q2: Baseline Focusing on Larger Timesteps**
> > > > >
> > > > > Thanks for the comment. We provide additional experiments that focus on larger timesteps.
> > > > >
> > > > > **Setting:**
> > > > >
> > > > > - We sample $\sigma = e^ z$ with $z\sim N(\mu_z, 1)$. Specifically, we set $\mu_ z=1$.
> > > > > - We then map $\sigma$ to the timestep domain via $t=\frac{\sigma}{1+\sigma }$.
> > > > >
> > > > > **Interpretation:**
> > > > >
> > > > > - The parameterization **puts more sampling weight on larger timesteps** than smaller ones.
> > > > >
> > > > > **Results:**
> > > > >
> > > > > | method                                                       | iteration | FID$\downarrow$ | sFID$\downarrow$ | IS$\uparrow$ |
> > > > > | ------------------------------------------------------------ | --------- | --------------- | ---------------- | ------------ |
> > > > > | SiT+log-normal($\mu_z$=0), **i.e., focusing on middle timesteps** | 50K       | 92.27           | **11.11**        | 14.76        |
> > > > > | SiT+log-normal($\mu_z$=1), **i.e., focusing on large timesteps** | 50K       | 91.89           | 22.97            | 14.43        |
> > > > > | SiT+REPA                                                     | 50K       | 78.20           | 11.71            | 17.10        |
> > > > > | SiT+HASTE                                                    | 50K       | **71.32**       | 13.85            | **19.26**    |
> > > > >
> > > > > **Conclusion:**
> > > > >
> > > > > - While **shifting the sampling weight to larger timesteps** achieves lower FID compared with original log-normal sampling, **acceleration methods based on external models** still achieve **better performance**.
> > > > > - This suggests that **pre-trained encoders provide valuable external semantic knowledge** to accelerate the training process.
> > > > >
> > > > > **Follow-up action:** We are going to add the comparison and discussion in the revision.
> > > > >
> > > > > Thanks again for your suggestion!
> > > > >
> > > > > > **Regarding Q3: Limitations of the Gradient-Angle Trigger**
> > > > >
> > > > > **Role of gradient signals:**
> > > > >
> > > > > - **Diagnostic tool:** The gradient analysis provides valuable diagnostic insights into **why and when** REPA transitions from helpful to harmful, which **motivates our investigation of termination strategies**.
> > > > > - **Termination guidance:** In addition to periodic performance monitoring during training, gradient analysis also **provides reference to identify the termination point**.
> > > > >
> > > > > While we acknowledge that we use the gradient signals mainly **for better understanding** in our current work, our work provides meaningful contributions:
> > > > >
> > > > > - **Diagnostic understanding:** Identification of **why and when** representation alignment becomes counterproductive.
> > > > > - **Practical solution:** We propose **a simple and effective solution** to tackle the mismatch in training process -- **alignment termination**, i.e., early stopping of regularization.
> > > > > - **Holistic alignment:** Integration of attention distillation with feature alignment shows **complementary benefits**.
> > > > > - **Empirical validation:** Consistent improvements across various architectures and datasets.
> > > > >
> > > > > **Future directions:** While our current gradient-based trigger has limitations (as reflected in A3 of Reviewer FRQ1), we continue to develop more robust and efficient automatic termination methods as a **promising direction for future work**.

---

### Official Review · Reviewer_qGU6 · 2025-06-30

**Clarity:** 4
**Significance:** 2
**Originality:** 3
**Rating:** 5
**Confidence:** 2

**Summary:**

This paper identifies a critical failure mode of Representation Alignment (REPA) in accelerating DiT training: as the DiT matures, REPA’s fixed teacher becomes a constraint. The authors diagnose this as a capacity mismatch via gradient analysis and propose HASTE, which combines holistic alignment (features + attention) with stage-wise termination of the alignment loss. This enables rapid early convergence and avoids late-stage degradation. Experiments on ImageNet show up to 28x acceleration over vanilla training.

**Questions:**

- Can you provide experiments validating gradient-angle-based automatic triggers?
- Could you test termination in text-to-image training to strengthen claims of generality?
- Do you have guidelines for τ selection without exhaustive tuning?
- for clarity, would it be possible to include pseudocode for the HASTE training loop?
- The ablation studies show that attention alignment (ATTA) by itself provides a significant acceleration, similar to REPA. In data-scarce or computationally constrained scenarios, would using ATTA alone be a more efficient and sufficient strategy than the full HASTE recipe?

**Ethical Concerns:**

["NO or VERY MINOR ethics concerns only"]

**Final Justification:**

My biggest concern was that the termination strategy wasn't tested on text-to-image models . The MS-COCO experiment successfully showed that the termination strategy works there too, which is a crucial addition as a result I'm happy to increase my score.

**Limitations:**

only thing is that the paper lacks discussion of risks from faster generative model training

**Quality:**

3

**Strengths And Weaknesses:**

**Strength**

- Insightful analysis of when REPA becomes counterproductive.
- HASTE dramatically accelerates DiT training on ImageNet
- Well-written
- The method is simple and implementable.

**Weakness*
- Optimal t requires manual tuning. While automatic triggers was suggested but not tested experimentally.
- Termination not tested on MS-COCO text-to-image tasks, undermining general claims of generalizability.
- Paper lacks discussion of risks from faster generative model training

---

> ### Author Rebuttal · Authors · 2025-07-31
>
> Thanks for the insightful and motivational comments from reviewer qGU6. We discuss the questions as follows.
>
> > **Q1:** Validating gradient-angle-based automatic trigger.
>
> **A1:** Thanks for the question. To address this concern, we provide a plain implementation as an example for an automatic gradient-angle trigger as follows.
>
> - **Setting:** We train SiT-B/2 on ImageNet 256 $\times$ 256 following the original settings of HASTE using a gradient-angle trigger.
> - **Implementation:**
>     - We start evaluating the gradient cosine similarity between holistic alignment and denoising objectives every 10K iterations from 50K iterations, when the training process has been relatively stable.
>     - In each round of evaluation, we compute the similarity separately on selected diffusion timesteps $t$ at different noise levels with 2048 images randomly sampled from ImageNet, and take the average value. When the average similarity falls below a threshold $\delta$, we terminate the holistic alignment.
>     - In the plain example, we perform gradient evaluation on the 5th SiT block (counting from 1). We select $t=0.02$, $t=0.1$, $t=0.5$ and $t=0.8$, and set $\delta=-0.05$.
> - **Results:**
>     - **Gradient cosine similarity:**
>
>         |timestep\iteration|50K|60K|70K|80K|
>         |-|-|-|-|-
>         |0.02|-0.0174|0.0418|0.0872|-0.0032|
>         |0.1|0.0126|-0.0767|-0.0148|-0.1155|
>         |0.5|0.0134|0.0106|-0.0848|-0.1272|
>         |0.8|0.0034|0.0406|0.1025|-0.0212|
>         |average|0.0030|0.0041|0.0225|**-0.0668(terminate)**|
>     - **Performance:**
>         |method|iteration|termination|FID$\downarrow$|sFID$\downarrow$|IS$\uparrow$|
>         |-|-|-|-|-|-|
>         |SiT|100K|-|63.46|7.31|20.6|
>         |SiT+HASTE|100K|-|39.86|7.16|35.8|
>         |SiT+HASTE|100K|80K|**38.69**|**6.88**|**36.4**|
>
> - **Findings:** Our automatic trigger terminates the holistic alignment at 80K iterations, and achieves better performance at 100K iterations.
>
> - **Limitations:** Though we validate the automatic gradient-angle trigger on SiT-B/2, we find it challenging to design such a trigger for precise termination, because of:
>     - **Additional overhead:** Gradient evaluation introduces extra computational cost, limiting the practical evaluation frequency.
>     - **Hyperparameter setting:** The automatic trigger is sensitive to hyperparameters, such as block choice, timestep selection, and the threshold.
>
> - **Conclusion:** Given that monitoring model performance during training is a standard practice in modern deep learning workflows, we think **it's more practical to integrate the periodic evaluation of both metrics and gradient cosine similarity to determine the optimal termination point**.
>
>
> > **Q2:** Validation of termination strategy in HASTE on MS-COCO.
>
> **A2:** To address this concern, we conduct the following text-to-image (T2I) generation experiment on MS-COCO.
>
> - **Setting**: Following the original settings of HASTE and REPA on MS-COCO, we continue to train MMDiT with both methods from 150K to 250K iterations, and compare the performance of MMDiT+HASTE with and without termination $\tau$=200K. As training progresses, we raise the classifier-free guidance (cfg) [1] scale to 3.5 for all evaluations at 250K iterations to steer the sampling process within the data manifold. We consistently use the SDE Euler-Maruyama sampler with NFEs=250.
> - **Results**:
>     |method|iteration|termination|FID$\downarrow$|
>     |-|-|-|-|
>     |MMDiT+REPA|250K|-|4.28|
>     |MMDiT+HASTE|250K|-|4.10|
>     |MMDiT+HASTE|250K|200K|**4.06**|
> - **Conclusion:** As expected, while our holistic alignment outperforms REPA for MMDiT, the termination strategy also leads to better performance in the T2I task.
>
> > **Q3:** Guidelines for τ selection.
>
> **A3:** Thanks for the question. We first explain our selection strategy in practice, and then give guidelines for τ selection.
> - **"Exhaustive" tuning is not necessary:**
>     - **Gradient analysis helps:** In practice, we don't really exhaustively tune $\tau$, because the gradient analysis helps us to locate the termination point. As we state in Lines 283-287, we choose to stop around 250K iterations when we observe **limited performance improvement** and **relatively low gradient cosine similarities**.
>     - **Tolerance of $\tau:$** Additionally, we find HASTE is relatively robust to $\tau$. In Figure 7b, we show that varying $\tau$ from 250K to 290K doesn't lead to a significant difference in the FID of SiT-XL/2 at 400K iteration.
> - **Guidelines for τ selection:** As we have mentioned above, the periodic evaluation of metrics and gradient cosine similarity helps to locate the approximate termination period. Additionally, termination within such a period works reliably well for HASTE and is more practical than locating the precise point.
>
> > **Q4:** Pseudocode for HASTE training loop.
>
> Thanks for the advice. We will add it to our latest version.
>
> > **Q5:** Using ATTA alone in data-scarce or computationally constrained scenarios.
>
> **A5:** Thank you for the insightful advice. In the following, we provide experiments on MS-COCO to explore the performance of ATTA, HASTE, and REPA in the given scenario.
>
> - **Setting:** To simulate data-scarce or computationally constrained conditions, we sample **20K** images ($<\frac{1}{4}$ original size) from MS-COCO training dataset, and set the batchsize of **64** ($\frac{1}{4}$ default batchsize), and train MMDiT for only **50K** iterations. We do not apply termination for HASTE.
> - **Results:**
>     |method|iteration|FID|
>     |-|-|-|
>     |MMDiT|50K|49.03|
>     |MMDiT+REPA|50K|32.59|
>     |MMDiT+ATTA|50K|43.17|
>     |MMDiT+HASTE|50K|**31.39**|
> - **Findings:** Under the specific setting, all three methods provide considerable acceleration to the training process. HASTE works best and REPA performs better than ATTA.
> - **Analysis:** As we mentioned in **Lines 167-169**, ATTA nails routing but lacks supervision over condition processing, which is even more evident in MMDiT architecture: we only perform attention alignment with the $QK^T$ matrix generated from the input image to avoid affecting textual processing. In contrast, REPA performs alignment over the hidden states of MMDiT, acting on both vision and textual processing. Therefore, combining feature and attention alignment provides more holistic guidance for training.
> - **Conclusion:** In data-scarce or computationally constrained scenarios, ATTA can accelerate the training process, but not as effectively as REPA and HASTE.
>
> > **Q6:** Discussion of risks from faster generative model training.
>
> **A6:** Thanks for the concern. We acknowledge that faster generative model training could potentially **lower barriers for harmful content generation** and **accelerate deepfake production**.
>
> While we recognize these concerns, we believe HASTE's risk profile is relatively contained for several reasons:
> - **Research-focused contribution:** Our method primarily benefits legitimate research by **reducing computational barriers**, enabling smaller labs and researchers to participate in generative modeling research.
> - **Safeguards applicable:** Standard safety measures, such as content filtering, ethical guidelines, and responsible disclosure, remain **compatible** with HASTE-trained models.
>
> **Limitations and future work:** As a technical contribution focused on training efficiency, our work doesn't directly address broader AI safety challenges, which we acknowledge as important future work. We are going to include safety concerns in Section 5 in the revision.

---

> > ### Comment · Reviewer_qGU6 · 2025-08-05
> >
> > Thank you for your detailed response. I appreciate the effort in adding new experiments and providing clarifications.
> > I am happy to increase my score, regardless I have the following questions that you could include as discussion points in the final version.
> >
> > - It's great to see that the termination strategy is beneficial for MS-COCO. I was just wondering why you raise the classifier-free guidance to 3.5?
> >
> > - I understand that REPA acts on both vision and text, while ATTA guides only the vision part. Now, do you think that there could be value in aligning not just the self-attention of the image encoder, but also the cross-attention maps that integrate text conditioning?

---

> > > ### Author Response · Authors · 2025-08-06
> > >
> > > Thank you for your positive feedback and insightful questions. We will address the questions and add relevant contents in the revision.
> > >
> > > > **Regarding Q2: CFG Scale Choice**
> > >
> > > **Theoretical background:**
> > >
> > > - **Training objective:** As a likelihood-based generative approach, the denoising objective of diffusion models is **equivalent to minimizing the forward Kullback–Leibler (KL) divergence** [3]. Forward KL is known for prioritizing **"mode-covering" behavior** and **forces the learned density to spread out**, potentially leading to blurry samples [2, 4].
> > > - **Need of guidance:** Therefore, diffusion models rely on **guidance techniques** [1, 2] in order to pull the sampling trajectory back to **the core of the data manifold**.
> > >
> > > **Practical consideration:**
> > >
> > > - **Observation:** In our experiments on **MS-COCO**, which is **a relatively small dataset** compared to larger-scale text-to-image benchmarks, we observe that the "spreading" problem becomes **increasingly evident** during extended training beyond 150K iterations.
> > > - **Remediation:** To **mitigate this tendency** and **ensure fair evaluation**, we increase the CFG scale to 3.5 during sampling **across all methods**.
> > >
> > > [1]. Ho & Salimans, *Classifier-Free Diffusion Guidance*, NeurIPS 2021.
> > >
> > > [2]. Karras et al., *Guiding a diffusion model with a bad version of itself*, NeurIPS 2024.
> > >
> > > [3]. Song et al., *Maximum Likelihood Training of Score-Based Diffusion Models*,  NeurIPS 2021.
> > >
> > > [4]. Zheng et al., *Direct Discriminative Optimization: Your Likelihood-Based Visual Generative Model is Secretly a GAN Discriminator*,  ICML 2025.
> > >
> > > > **Regarding Q5: Alignment for Cross-attention**
> > >
> > > Thanks for your suggestion. We implement ATTA on vision self-attention for MM-DiT out of several practical considerations:
> > >
> > > - **Fair comparison:** In our experiments on MS-COCO, we adopt MM-DiT architecture and the pre-trained DINOv2-B **following REPA to ensure fair comparison**.
> > > - **Architecture limitation:** Since DINOv2-B uses original ViT architecture with **mainly self-attention mechanism** for vision processing [1], we perform ATTA **mainly on vision attention maps** rather than the cross-attention of MM-DiT.
> > >
> > > **Future direction:** We sincerely appreciate your insight of alignment for cross-attention again. We will follow the suggestion to further explore attention alignment over **cross-attention** maps between MM-DiT and pre-trained **multi-modal** encoders, and constantly improve our manuscript.
> > >
> > >
> > > [1]. Oquab et al., *DINOv2: Learning Robust Visual Features without Supervision*, TMLR 2024.
> > >
> > > Thanks for your acknowledgement of our analysis and effort!

---

> > > > ### Comment · Reviewer_qGU6 · 2025-08-07
> > > >
> > > > Thank you for the clarifications. I highly recommend to include these discussions in the final revision.

---

> > > > > ### Author Response · Authors · 2025-08-08
> > > > >
> > > > > Thanks for your time and attention. We are going to add the discussions to the revision, and welcome any further questions.

---

### Official Review · Reviewer_fiaM · 2025-07-02

**Clarity:** 3
**Significance:** 2
**Originality:** 2
**Rating:** 4
**Confidence:** 4

**Summary:**

This paper proposes improvements to REPA, a method for accelerating the training of diffusion models. Two key enhancements are introduced:

- **Termination** $\tau$: Instead of applying REPA throughout the entire training, it is stopped partway through.

- **ATTA**: Beyond aligning features, an additional alignment loss encourages the student model’s attention maps to resemble those of the vision encoder.

These two strategies yield improvements over vanilla REPA in terms of both training efficiency and performance.

**Questions:**

- Could you clarify precisely what you mean by “capacity mismatch”? How does this relate to the divergence between gradient signals?

- In ATTA, the CLIP attention quality seems degraded by attention sinks. Would it be possible to use register tokens to remove these sinks from the attention maps before computing ATTA? Recent work has proposed applying register tokens without additional training [1]

[1]: Vision Transformers Don't Need Trained Registers (https://arxiv.org/abs/2506.08010)

**Ethical Concerns:**

["NO or VERY MINOR ethics concerns only"]

**Final Justification:**

I believe this paper suggests a simple but effective strategy for improving REPA. However, because the REPA is just a single regularization technique for training the diffusion model, I think the impact is limited. Therefore I suggest BA.

**Limitations:**

The paper clearly discusses limitations related to model architecture, pixel-level diffusion, and video generation, which is appreciated.

**Quality:**

3

**Strengths And Weaknesses:**

**Strengths**

- The writing is very clear and accessible, making the paper easy to follow.

- The proposed solution is simple yet effective, achieving substantial gains.

- The ablation studies clearly demonstrate how each component contributes to improving REPA, with Table 3 particularly compelling.

**Weaknesses**

- The approach heavily relies on heuristics. For example, while the authors attribute the necessity of termination to a “capacity mismatch,” the evidence in Section 2.2 does not strongly support this explanation. It might have been more straightforward to describe this simply as gradient signal interference.

- Additionally, I find Figures 3 and 4 somewhat disappointing and potentially misleading. In Appendix A, the cosine similarity plots show that the gradient signals only begin to point in opposite directions after much longer training (4M iterations), rather than within the training regime actually used (500K iterations).

---

> ### Author Rebuttal · Authors · 2025-07-31
>
> Thanks reviewer fiaM for adoring our work. We are delighted to discuss the questions below.
>
> > **W1&Q1:** Relationship between "capacity mismatch" and "gradient divergence".
> Thanks for your insightful question. We first provide a further detailed clarification of "capacity mismatch", and then explain how it connects with "gradient divergence".
>
> - **Capacity Mismatch:** We refer "capacity mismatch" to two **fundamental** differences between vision representation models (e.g., DINOv2) and diffusion models.
>     - **Task mismatch:** Diffusion models are designed to generate high-quality samples through denoising, while representation models are aimed at capturing semantic information and constructing good representations. Therefore, the two kinds of models possess fundamentally different objectives.
>     - **Distribution mismatch:** Diffusion models learn complex, noisy distributions across different noise levels. In contrast, representation models learn only through the clean marginal data distribution. In the training process, external representations help diffusion models to get a coarse and fast grasp of the generative task but fail to further improve the fine-grained generation capacity, especially at diffusion timesteps with lower noise levels.
> - **Relationship**:
>     - **Statement concern:** While we acknowledge that gradient interference is a more direct description, we would like to clarify that capacity mismatch serves as the cause and provides deeper insight, which helps to explain the source of training constraint. We also follow that insight to design the low-frequency filter experiment in Figure 5, Section 2.2.
>     - **Experimental evidence:** In Section 2.2, we leverage gradient analysis for diagnosis and observe the Ignition-Plateau-Conflict trends, which are consistent with the shift of REPA from accelerator to brake. Additionally, our low-frequency filter experiment further verifies the existence of capacity mismatch.
>     - **Mismatch and divergence:** When diffusion models seek high-frequency detail to further improve the generation quality, the capacity mismatch leads to the constraint of improvement, which is obvious at diffusion timesteps at lower noise levels (e.g., 0-0.1). At these timesteps, we observe evident gradient divergence.
>
> > **W2:** Mismatch between gradient direction figures and cosine similarity plots.
>
> Thanks for your careful observation. We understand the concern about potential inconsistency between the main text figures and the appendix data. Please allow us to clarify this important point.
>
> - **Timestep-dependent gradient evolution:**
>
>   As reported in Appendix A.1, we observe that gradient cosine similarities across different diffusion timesteps exhibit similar overall trends—starting relatively high, gradually decreasing, and eventually becoming negative. However, **the rate of this evolution varies significantly across timesteps**:
>
>   - For t ∈ (0, 0.1] (fine detail generation), we observe negative cosine within 500K iterations.
>   - For larger timesteps (coarser features), similar behavior appears much later.
>   - Figures 3 and 4 are **schematic illustrations** intended to demonstrate this general trend, with line 98 explicitly noting that Figure 3 uses t = 0.1 as an example.
>
> - **Balancing alignment benefits and conflicts**
>
>   Our method fundamentally seeks to balance two competing effects:
>
>   - **Benefits**: Alignment provides valuable guidance for low-frequency, structural features
>   - **Costs**: Alignment eventually hinders the learning of high-frequency details
>
>   The key is to **terminate before the harm to fine-grained details outweighs the benefits to coarse-grained structure**. Notably, our actual termination point (250K iterations for HASTE) is chosen well before gradient similarities become fully negative across all timesteps. This reflects a principled trade-off rather than waiting for complete gradient opposition. Additionally, our experimental results with HASTE strongly validate this termination strategy.
>
> > **Q2:** Using register tokens to remove attention sinks in CLIP.
>
> Thanks for your advice. First, we would like to clarify that our original submission doesn't include experiments of ATTA with CLIP attention. Nevertheless, we can provide the experimental results of HASTE with CLIP-ViT-B/16.
>
> - **Setting:** We use SiT-B/2 for fast validation following the original settings of HASTE. Since SiT-B/2 and CLIP-ViT-B/16 possess different patch numbers (i.e., 16x16 and 14x14), we perform additional spatial interpolations to align the dimensions.
>
> - **Results (w/o cfg):**
>
>   - Comparing HASTE with REPA at 100K iteration:
>
>     | method    | iteration | encoder       | FID↓      | sFID↓    | IS↑      |
>     | - | - | - | - | - | - |
>     | SiT       | 100K      | -             | 63.46     | 7.31     | 20.6     |
>     | SiT+REPA  | 100K      | DINOv2-B      | 49.50     | **7.00** | 27.5     |
>     | SiT+REPA  | 100K      | CLIP-ViT-B/16 | 54.92     | 7.63     | 24.7     |
>     | SiT+HASTE | 100K      | DINOv2-B      | **39.86** | 7.16     | **35.8** |
>     | SiT+HASTE | 100K      | CLIP-ViT-B/16 | 48.74     | 7.92     | 28.7     |
>
>   - Validating the termination strategy of HASTE:
>
>     | method    | iteration | encoder       | termination | FID↓      | sFID↓    | IS↑      |
>     | - | - | - | - | - | - | - |
>     | SiT       | 150K      | -             | -           | 52.71     | 7.06     | 26.2     |
>     | SiT+HASTE | 150K      | CLIP-ViT-B/16 | -           | 38.36     | 7.23     | 38.2     |
>     | SiT+HASTE | 150K      | CLIP-ViT-B/16 | 100K        | **36.13** | **6.95** | **40.5** |
>
> - **Register tokens:** We sincerely appreciate the insightful paper you referred to. We think it's a promising direction to leverage the register tokens without additional training to improve the attention quality of pre-trained encoders. We will discuss the possibility introduced by the proposed method in **Limitations and future work**,  Section 5.

---

> > ### Comment · Reviewer_fiaM · 2025-08-05
> >
> > > **W1 & Q1: Capacity mismatch**
> >
> > This is just my personal opinion, but if "capacity mismatch" is intended to capture the idea of *task mismatch + distribution mismatch*, then the term might be a bit misleading. Just from the name, it sounds like it's referring to issues caused by differences in model parameter sizes. It would be nice if a more precise term could be used instead.
> >
> > > *Distribution mismatch: Diffusion models learn complex, noisy distributions across different noise levels. In contrast, representation models learn only through the clean marginal data distribution.*
> >
> > If this explanation is based on that intuition, then wouldn’t we expect **less** gradient divergence at small timesteps, where the input is closer to being clean and thus more similar to the representation model's domain? I do agree at a high level that there is a mismatch between the learning objectives of representation models and generative models, but when trying to specify the root cause, there are some points that feel a bit counterintuitive. That’s why I suggested using a more neutral and direct term like **gradient divergence**, rather than a more interpretive term like *mismatch*.
> >
> > To be clear, I’m not insisting this should be changed—please just take this as one reader’s perspective.
> >
> > > **W2: Mismatch between gradient direction figures and cosine similarity plots**
> >
> > Yes, I agree that the paper is largely empirical, and that the illustrations are schematic in nature. My concern with both W1 and W2 is actually quite similar: I’m still not fully sure **why** the method works so well. At a high level, the idea that there’s a mismatch between the representation and generative models makes sense, but that seems to be the furthest the explanation goes.
> >
> > > **Q2: Using register tokens to remove attention sinks in CLIP**
> >
> > The point about CLIP was likely a misunderstanding on my part. Thank you for running the additional experiments nonetheless. The fact that DINO performs better than CLIP in terms of FID is quite understandable.

---

> > > ### Author Response · Authors · 2025-08-05
> > >
> > > Thanks for your insightful follow-up regarding the "capacity mismatch". We are delighted to provide more explanations in detail about the concept.
> > >
> > > > **Regarding W1 & Q1: Divergence at Small Timesteps**
> > >
> > > - **Experimental phenomenon:**
> > >   - While intuitively one might expect smaller divergence at small timesteps, **our gradient analysis reveals a counterintuitive pattern**: the cosine similarity between alignment and denoising objectives is actually **highest at intermediate timesteps**, rather than small timesteps.
> > >   - This phenomenon **aligns with findings from REPA's** Appendix C.2 (DiT Analysis) [1], where they similarly observe that **a pretrained DiT's representations align better with representation models at intermediate timesteps** than at smaller ones.
> > >
> > > - **Our explanation:**
> > >   - **Diffusion models** learn and generate through a **differential approach**, estimating the **score function** $\nabla_x logp_t(x)$ of the data distribution. **At very small timesteps**, though the input image is relatively clean, the diffusion model's task is to **continue denoising to achieve better quality**. We believe that at this noise level, diffusion models develop representations **specialized for fine-grained detail generation**.
> > >   - **Representation models**, in contrast, learn **directly from the clean data distribution**, accumulating more **general-purpose representations** suitable for downstream tasks.
> > >   - Therefore, **aligning representations at small timesteps during late training stages becomes a constraint on diffusion training**, which we aim to capture with the term "capacity mismatch."
> > >
> > >
> > > [1]. Yu et al., *Representation Alignment for Generation: Training Diffusion Transformers Is Easier Than You Think*, ICLR 2025.
> > >
> > > > **Regarding W2:** **Positive Performance of HASTE**
> > >
> > > We acknowledge that we can identify more **specific mechanisms** beyond the high-level mismatch:
> > >
> > > - **Progressive specialization:** Early in training, **both objectives** point toward **a coarse grasp** of semantic structure. As training progresses, the diffusion objective **increasingly demands fine-grained generation capabilities** that conflict with the representation model's invariant features.
> > > - **Empirical validation:** The validation of our termination strategy **across different model sizes and datasets** suggests this is a robust phenomenon, even if our **mechanistic understanding** could be deeper.
> > >
> > > **Reflection:** We sincerely appreciate your advice from **the reader’s perspective**. Your thoughtful comments are rather helpful for us to improve our mechanistic explanation.

---

### Official Review · Reviewer_KpE1 · 2025-07-03

**Clarity:** 3
**Significance:** 2
**Originality:** 2
**Rating:** 4
**Confidence:** 4

**Summary:**

This paper addresses the slow training of Diffusion Transformers (DiTs) by improving Representation Alignment (REPA). The authors observe that REPA accelerates early training but hinders performance later due to a "capacity mismatch" between the generative student model and the non-generative teacher (e.g., DINOv2). They propose HASTE (Holistic Alignment with Stage-wise Termination), a two-phase approach, which achieves up to 28× faster training on ImageNet and improves text-to-image generation on MS-COCO, matching or surpassing REPA without architectural changes. The method combines gradient-angle analysis to diagnose alignment conflicts and stage-wise termination to avoid over-regularization.

**Questions:**

Please refer to the weakness part

**Ethical Concerns:**

["NO or VERY MINOR ethics concerns only"]

**Final Justification:**

Thanks for the authors' reply. I keep my original rating.

**Limitations:**

Please refer to the weakness part

**Quality:**

3

**Strengths And Weaknesses:**

Strength:
- The authors identify the capacity mismatch between generative and non-generative models, supported by gradient-angle analysis showing how REPA shifts from helpful to harmful .
- Integrating attention maps (relational priors) with feature alignment (semantic anchors) provides a more comprehensive guidance mechanism .
- Extensive experiments on ImageNet and MS-COCO demonstrate significant speed-ups and performance gains, with ablation studies supporting the effectiveness of each component .

Weakness:
- The paper primarily uses fixed iterations for termination (e.g., τ=100K or 250K) but mentions gradient-angle thresholds without detailing how they are practically implemented or optimized .
- While attention maps are shown to improve training, the paper lacks a theoretical framework explaining why mid-layer attention distillation is optimal, especially for diffusion models .
- On MS-COCO, HASTE does not apply termination due to "limited iterations," leaving unanswered questions about its effectiveness in text-guided generation over longer runs .
- Combining feature and attention alignment may introduce additional training costs, which the paper does not explicitly quantify (e.g., memory or FLOPs) .
- The choice of mid-layer alignment (e.g., SiT blocks 4–7) is empirical, but the paper does not systematically explore how layer depth affects performance .
- Methods like TREAD [23] or FasterDiT [49] are cited but not directly compared in all experiments, limiting the scope of the performance claims .
- The paper uses DINOv2 as the teacher, but it does not analyze how different pre-trained encoders (e.g., CLIP or Swin Transformer) affect HASTE’s performance .
- While FID improvements are highlighted, the paper does not address scenarios where HASTE might underperform, such as when generating highly complex or rare visual concepts .

---

> ### Author Rebuttal · Authors · 2025-07-31
>
> Thanks reviewer KpE1 for constructive comments. We are delighted to discuss the questions below and sincerely hope these answers can address the concerns.
>
> > **Q1:** Implementation of gradient-angle threshold.
>
> **A1:** Thanks for your question. Here’s a clarification and an updated revision for transparency.
>
> - **Dual termination strategy:** We mainly use fixed iteration $\tau$ due to its simplicity and effectiveness (*e.g.*, ImageNet/MS-COCO). The gradient-angle threshold adds theoretical benefits. Gradient analysis (Lines 283–287) shows alignment gains drop around **250K iterations**.
>
> - **Gradient-angle trigger (plain implementation):** We train SiT-B/2 on ImageNet 256 $\times$ 256 following the original settings of HASTE using a gradient-angle trigger.
>
>     - **Process:**
>         - After 50K iterations, every 10K, we check the gradient cosine similarity between alignment and denoising objectives.
>
>         - At each check, we use selected timesteps $t$ at various noise levels on 2048 random ImageNet images; average the resulting similarities. If average $<$ threshold $\delta$, we stop alignment.
>
>         - Example uses 5th SiT block, timesteps $t=0.02$, $0.1$, $0.5$, $0.8$, and $\delta=-0.05$.
>
>     - **Results:**
>         - **Cosine similarity:**
>
>         |timestep\iteration|50K|60K|70K|80K|
>         |-|-|-|-|-|
>         |0.02|-0.0174|0.0418|0.0872|-0.0032|
>         |0.1|0.0126|-0.0767|-0.0148|-0.1155|
>         |0.5|0.0134|0.0106|-0.0848|-0.1272|
>         |0.8|0.0034|0.0406|0.1025|-0.0212|
>         |avg|0.0030|0.0041|0.0225|**-0.0668(terminate)**|
>
>         - **Performance:**
>
>         |method|iter|terminate|FID$\downarrow$|sFID$\downarrow$|IS$\uparrow$|
>         |-|-|-|-|-|-|
>         |SiT|100K|-|63.46|7.31|20.6|
>         |SiT+HASTE|100K|-|39.86|7.16|35.8|
>         |SiT+HASTE|100K|80K|**38.69**|**6.88**|**36.4**|
>
>     - **Key point:** The automatic trigger ends holistic alignment at 80K and improves results at 100K.
>
>     - **Limitations:**
>         - **Overhead:** Gradient checks add computation, limiting check frequency.
>         - **Hyperparameters:** Trigger is sensitive to hyperparameters (block, timestep, threshold).
>
>     - **Conclusion:** Since monitoring model performance during training is a standard practice in modern deep learning workflows, **it's more practical to integrate the periodic evaluation of both metrics and gradient cosine similarity to determine the optimal termination point**.
>
>
> > **Q2&Q5:** Exploration of alignment layer depth and theoretical framework for mid-layer attention distillation.
>
> **A2&A5:** Thanks for your question. Below are concise explorations of alignment depth and our choice of mid-layer for distillation.
>
> - **Alignment depth:**
>     - **Setup:** Train SiT-B/2 on ImageNet 256$\times$256 with HASTE for 100K iters. **Vary alignment depth; no termination.**
>
>     - **Results:**
>         |method|DINO|SiT|Depth|FID$\downarrow$|sFID$\downarrow$|IS$\uparrow$|
>         |-|-|-|-|-|-|-|
>         |SiT|[7,9,11]|[2,3,4]|5|**39.86**|**7.16**|**35.8**|
>         |SiT+HASTE|[7,9,11]|[4,5,6]|7|40.59|7.27|35.6|
>         |SiT+HASTE|[7,9,11]|[6,7,8]|9|43.37|7.75|33.5|
>
>     - **Conclusion:** Depth **5** yields best performance.
>
> - **Mid-layer distill choice:**
>
>     - Alignment depth set per REPA (e.g., 8 for SiT-L/XL), to observe ATTA for HASTE.
>
>     - As in Lines 171-183, mid-layer choice is due to:
>         - **Shallow mismatch:** Early layers should handle noisy input. (SiT-L/2: layer 4-7 > layer 2-7 in Table 6)
>         - **Deep freedom:** Deeper layers focus on denoising. Aligning too deep hurts results (REPA: depth >8 reduces performance). So, use layer 4-7 for alignment.
>
> - **Limits & future work:** While empirical layer choice works, theory may explain more. We plan to study attention in diffusion transformers further.
>
>
> > **Q3:** Validation of termination strategy in HASTE on MS-COCO.
>
> **A3:** To address your concern, we ran a T2I test on MS-COCO.
>
> - **Setup**: Using HASTE/REPA settings, we trained MMDiT with both from 150K-250K iterations, comparing MMDiT+HASTE with/without termination at $\tau$=200K. At 250K, cfg set to 3.5 for eval; Sampler: SDE Euler-Maruyama, NFEs=250.
>
> - **Results**:
>
>     |method|iter|term|FID$\downarrow$|
>     |-|-|-|-|
>     |MMDiT+REPA|250K|-|4.28|
>     |MMDiT+HASTE|250K|-|4.10|
>     |MMDiT+HASTE|250K|200K|**4.06**|
>
> - **Conclusion:** Holistic alignment beats REPA for MMDiT; termination strategy further improves T2I.
>
>
> > **Q4:** Quantification of additional training costs introduced by holistic alignment.
>
> **A4:**
>
> Thank you for the question. We evaluate the extra overhead of HASTE on SiT-XL/2 introduced by external vision model DINOv2-B:
>
> - **Extra memory overhead:** The peak CUDA memory with a local batchsize 32:
>
>     |HASTE|REPA|
>     |-|-|
>     | 30666.84 MB | 27818.01 MB |
>
> - **DINOv2-B forward overhead:** During training, we compute DINOv2‑B representations online following REPA. The forward pass of DINOv2‑B incurs approximately 23.19 GFLOPs for the base‑size model at 224×224 input resolution without XFormers.
>
> > **Q6**: Comparison with TREAD and FasterDiT.
>
> **A6:** Thanks for your comment. Here’s a detailed comparison with TREAD[1] and FasterDiT[2].
>
> - **ImageNet 256$\times$256**
>   **Since TREAD does not report SiT+TREAD+cfg, we use their best DiT+TREAD result.**
>
>     - **Results:**
>
>       - w/o cfg
>
>         |method|epoch|FID$\downarrow$|
>         |-|-|-|
>         |FasterDiT|400|7.91|
>         |SiT+HASTE|80|7.31|
>         |SiT+TREAD|80|**4.89**|
>
>       - w/ cfg
>
>         |method|epoch|FID$\downarrow$|
>         |-|-|-|
>         |FasterDiT|400|2.03|
>         |DiT+TREAD|740|1.69|
>         |SiT+HASTE|400|**1.44**|
>
>    - **Analysis:** TREAD gets lower FID w/o cfg, but **HASTE is better w/ cfg**. TREAD notes, “Restricting info in training can hurt CFG performance.” Our HASTE gains a lot with cfg.
>
>      - HASTE at 600 epochs even outperforms REPA’s best FID:
>
>        |method|epoch|FID$\downarrow$|sFID$\downarrow$|IS$\uparrow$|
>        |-|-|-|-|-|
>        |SiT+REPA|800|1.42|4.70|**305.7**|
>        |SiT+HASTE|600|**1.41**|**4.51**|296.2|
>
>    - **Conclusion:** As CFG is key for most gen tasks, HASTE is **more practical** for acceleration.
>
> - **MS-COCO 256$\times$256**
>   We did not compare with FasterDiT or TREAD in T2I because:
>
>     - FasterDiT lacks T2I results/public code.
>
>     - Our model/setup differs from TREAD:
>
>       - TREAD uses smaller DiT-B, HASTE uses larger MMDiT.
>
>       - TREAD uses FID-10K, HASTE uses FID-50K.
>
>
> > **Q7: Exploration of different pre-trained encoders.**
>
> **A7:** Thanks for your advice. To address this, we further explore using CLIP(ViT-B/16) as the pre-trained encoder:
>
> - **Setting:** We use SiT-B/2 for fast val as in HASTE's original setup. Since SiT-B/2 and CLIP-ViT-B/16 have different patch numbers (16x16 vs. 14x14), we interpolate spatial dims for alignment:
>
>   - **Feature map interp**: SiT hidden states `[bs,16x16,768]` are reshaped to `[bs,16,16,768]`, then downsampled with bilinear interp from `[16,16]` to `[14,14]`.
>
>   - **Attn map interp**: SiT attention maps `[bs,H,16x16,16x16]` are reshaped to `[bs,H,16,16,16,16]` and interpolated on key/query spaces.
>
> - **Results (w/o cfg):**
>
>   - **HASTE vs REPA at 100K:**
>
>     |method|iter|encoder|FID↓|sFID↓|IS↑|
>     |-|-|-|-|-|-|
>     |SiT|100K|-|63.46|7.31|20.6|
>     |SiT+REPA|100K|DINOv2-B|49.50|7.00|27.5|
>     |SiT+REPA|100K|CLIP-ViT-B/16|54.92|7.63|24.7|
>     |SiT+HASTE|100K|DINOv2-B|39.86|7.16|35.8|
>     |SiT+HASTE|100K|CLIP-ViT-B/16|48.74|7.92|28.7|
>
>   - **HASTE termination at 150K:**
>
>     |method|iter|encoder|term|FID↓|sFID↓|IS↑|
>     |-|-|-|-|-|-|-|
>     |SiT|150K|-|-|52.71|7.06|26.2|
>     |SiT+HASTE|150K|CLIP-ViT-B/16|-|38.36|7.23|38.2|
>     |SiT+HASTE|150K|CLIP-ViT-B/16|100K|36.13|6.95|40.5|
>
> - **Analysis:** At 100K, both REPA and HASTE with CLIP-ViT-B/16 speed up SiT-B/2 training. DINOv2-B gives better gains, matching [REPA] that **semantically meaningful pre-trained encoders** enhance gen performance. At 150K, earlier termination with HASTE further lowers FID.
>
> - **Conclusion:** HASTE with CLIP-ViT-B/16 shows its generalizability across different encoders. We handle patch mismatch by interpolation, not just via resizing.
>
> In sum, Swin Transformer[3]'s window attn/patch merging means its features differ from those in diffusion transformers, so we use ViT-based encoders (e.g., DINOv2) for easier analysis and implementation; experiments focus on these.
>
>
> > **Q8: While FID improvements are highlighted, the paper does not address scenarios where HASTE might underperform, such as when generating highly complex or rare visual concepts.**
>
> **A8:** Thanks for your comment. First, we clarify that our work mainly focuses on:
>
> - relation between vision representative models & diffusion models,
>
> - and acceleration Methods for generative diffusion training,
>
> not on solutions for spec. hard cases. So, we test mainly on widely-used gen. datasets (e.g., ImageNet), and note HASTE may underperform in some conditions.
>
> We also evaluate HASTE in more complex cases:
>
> - **Text-to-image generation:** In our paper, we test HASTE on MS-COCO 256x256, where MMDiT captures complex semantics with ~80K train imgs.
>
> - **High-resolution image generation:** We also test HASTE on ImageNet 512x512. We set $\tau=300K$, cfg scale 1.25.
>
>   |method|iters|term.|FID$\downarrow$|sFID$\downarrow$|IS$\uparrow$|
>   |-|-|-|-|-|-|
>   |SiT+REPA|400K|-|2.55|**4.16**|241.2|
>   |SiT+REPA|500K|-|2.36|**4.16**|254.2|
>   |SiT+HASTE|400K|300K|2.49|4.20|231.4|
>   |SiT+HASTE|500K|300K|**2.34**|4.23|253.4|
>
> Thanks for your insightful question, we will explore rare visual concept generation in future work.
>
>
>
> [1] TREAD: Token Routing for Efficient Architecture-agnostic Diffusion Training. In arXiv.
>
> [2] FasterDiT: Towards Faster Diffusion Transformers Training without Architecture Modification. In NeurIPS 2024.
>
> [3] Swin Transformer: Hierarchical Vision Transformer using Shifted Windows. In ICCV 2021.

---

### Decision · Program_Chairs · 2025-09-17

**Decision:**

Accept (poster)

**Comment:**

This paper identifies the issues of current representation alignment for diffusion models with empirical analysis. Then the paper proposes an early stopping strategy for representation alignment. Additionally, the representation alignment is extended to hybrid attention alignment, and the alignment is only applied to certain blocks.

Reviewers commented positively on the identification of the capacity mismatch, the associated empirical studies, the enhanced alignment integrating attention, and extensive experiments.

Reviewers commented negatively on the naive strategy for termination using fixed iterations, the method design is mainly based on empirical and heuristic approaches, and the limited impact. Particularly, the gradient-angle trigger is only for analysis, not used as part of the method.

Since the insightful empirical analysis is the main contribution of this paper, not using the gradient-angle trigger does not undermine the main contribution. Therefore, it is recommended to accept. But the authors do need to carefully revise the related wording in the paper to avoid misunderstanding.